# Structure-dependence and metal-dependence on atomically dispersed Ir catalysts for efficient n-butane dehydrogenation

Xiaowen Chen[1,2,10], Xuetao Qin[3,10], Yueyue Jiao[4,5,6,10], Mi Peng[3], Jiangyong Diao[1], Pengju Ren[4,5], Chengyu Li[3], Dequan Xiao [7], Xiaodong Wen [4,5], Zheng Jiang [8], Ning Wang [9], Xiangbin Cai [9] ✉, Hongyang Liu [1,2] ✉ & Ding Ma [3] ✉

Single-site pincer-ligated iridium complexes exhibit the ability for C-H activation in homogeneous catalysis. However, instability and difficulty in catalyst recycling are inherent disadvantages of the homogeneous catalyst, limiting its development. Here, we report an atomically dispersed Ir catalyst as the bridge between homogeneous and heterogeneous catalysis, which displays an outstanding catalytic performance for n-butane dehydrogenation, with a remarkable n-butane reaction rate (8.8 mol·$g_{Ir}^{-1}$·$h^{-1}$) and high butene selectivity (95.6%) at low temperature (450 °C). Significantly, we correlate the BDH activity with the Ir species from nanoscale to sub-nanoscale, to reveal the nature of structure-dependence of catalyst. Moreover, we compare Ir single atoms with Pt single atoms and Pd single atoms for in-depth understanding the nature of metal-dependence at the atomic level. From experimental and theoretical calculations results, the isolated Ir site is suitable for both reactant adsorption/activation and product desorption. Its remarkable dehydrogenation capacity and moderate adsorption behavior are the key to the outstanding catalytic activity and selectivity.

Alkenes are the irreplaceable cornerstone for value-added chemicals in the modern chemical industry. Catalytic dehydrogenation of alkanes from natural gas is a direct and simple method to produce alkenes. In fundamental research, alkane dehydrogenation can be conducted with either homogeneous or heterogeneous catalysts. The homogeneous catalysis includes transfer dehydrogenation and acceptorless dehydrogenation, which have made significant progress in the last three decades[1–4]. In 1979, Crabtree and co-workers pioneered the study of stoichiometric transfer dehydrogenation of alkanes, mediated by a homogeneous iridium pincer complex. With tert-butylethylene as a hydrogen acceptor, the single-site iridium pincer complex could catalyze the dehydrogenation of cyclooctane and cyclopentane and form iridium complexes of cyclooctadiene and cyclopentadiene[5]. In 1999, Goldman and co-workers reported the first catalytic system for regioselective transfer dehydrogenation of linear alkanes to α-olefins[6], using pincer iridium complexes. Since then, various single-site pincer

[1]Shenyang National Laboratory for Materials Science, Institute of Metal Research, Chinese Academy of Sciences, Shenyang 110016, P. R. China. [2]School of Materials Science and Engineering, University of Science and Technology of China, Shenyang 110016, P. R. China. [3]Beijing National Laboratory for Molecular Sciences, College of Chemistry and Molecular Engineering, Peking University, Beijing 100871, P. R. China. [4]State Key Laboratory of Coal Conversion, Institute Coal Chemistry, Chinese Academy of Sciences, Taiyuan 030001, P. R. China. [5]National Energy Center for Coal to Clean Fuel, Synfuels China Co., Ltd, Beijing 100871, P. R. China. [6]The University of Chinese Academy of Sciences, Beijing 100049, P.R. China. [7]Center for Integrative Materials Discovery, Department of Chemistry and Chemical and Biomedical Engineering, University of New Haven, West Haven, CT 06516, USA. [8]Shanghai Institute of Applied Physics, Chinese Academy of Sciences, Shanghai 201204, P. R. China. [9]Department of Physics and Center for Quantum Materials, Hong Kong University of Science and Technology, Kowloon, Hong Kong SAR, P. R. China. [10]These authors contributed equally: Xiaowen Chen, Xuetao Qin, Yueyue Jiao.
✉e-mail: xcaiak@connect.ust.hk; liuhy@imr.ac.cn; dma@pku.edu.cn

iridium complexes have been developed. Based on a (PCP)Ir coordination structure, the steric and electronic properties of pincer iridium complex can be precisely tuned by (1) substituting functional groups on two phosphorus atoms, (2) altering linkers between the backbone and two P atoms, (3) modifying para-position on the aromatic backbones or substituting other aromatic hydrocarbon backbones, and (4) introducing other metal atoms. The decreased steric hindrance in this type of catalyst favors catalytic activity and accelerates β-H elimination with regioselectivity to form terminal or internal olefin[7]. Moreover, strong linker-atom to C(aryl) π-donation can facilitate the rate-determining step in alkane C−H bond addition to 14e (pincer)Ir fragment, and thus enhance catalytic activity in the transfer dehydrogenation reaction[8]. However, limited by thermal stability, the highly efficient pincer iridium complexes suffered from decomposition under slightly harsh conditions (>300 °C). The difficulty of catalyst separation and recycling restricted large-scale industrial applications. To overcome the limitations, one solution is to immobilize iridium pincer complexes on oxide supports through covalent bonding to anchor Ir or para-position atoms. This strategy can be considered as the transformation of homogeneous catalysis to surface organometallic catalysis[2,9–11]. The highest stable temperature of single-site iridium pincer complexes on the supported oxides was above 300 °C, which was 100 °C higher than other homogeneous complexes with similar structures. But the collapse of the pincer ligand and loss of the Ir oxidation state caused fast and inevitable deactivation at >350 °C in heterogeneous alkane dehydrogenation[9]. Therefore, fabricating thermally stable isolated Ir atom catalysts to replace single-site iridium pincer complexes remains a challenge for alkane dehydrogenation.

For heterogeneous alkane dehydrogenation, C−H cleavage is active on noble metals such as Rh, Ru, Os, Ir, Pt, and Pd, which preferentially bind with −CH₃, considered as carbon-preferred transition metals[12,13]. Previously, Ir nanoparticles (NPs) dispersed on oxide supports were designed for propane dehydrogenation (PDH). A second metal Sn was introduced as the promoter, which can separate larger Ir ensembles into highly dispersed and uniform Ir NPs and simultaneously modify the electronic property of the catalyst by forming IrSn alloy[14,15]. However, the Ir active species in the nanoscale has non-ideal atomic efficiency, resulting in poor activity in dehydrogenation reactions when normalized to Ir. In order to maximize the atomic efficiency, single-atom catalysts (SACs) have been designed for the alkane dehydrogenation process, such as nitrogen-doped carbon-supported Ru single atoms (SAs)[16], γ-Al₂O₃ supported Pt₁Cu single-atom alloy (SAA)[17], SiO₂ supported Rh₁Cu SAA[18], MFI siliceous zeolite confined single Fe sites[19], SiO₂ confined single Co sites[20], and tetrahedral Co(II) sites[21]. For example, Zhang et al. designed the highly stable Ru SACs for PDH[16]. The Ru species remained almost unchanged compared with the fresh catalyst, even after reduction treatment by H₂ at 600 °C or the PDH reactions, indicating that Ru species remained atomically dispersed and has partially oxidized state during PDH. Similarly, Jeffrey T. Miller et al. developed an γ-Al₂O₃ supported isolated Ni (II) site by anchoring Ni²⁺ cations into Al³⁺ vacancy on γ-Al₂O₃ as a catalyst for PDH[22]. The nature of the Ni sites remains constant for the fresh sample, regardless of 1 h treatment by 3% H₂ at 600 °C or 3% C₃H₈ at 580 °C or 20% O₂ at 600 °C, suggesting that the atomically dispersed Ni²⁺ sites retain the local structure over reduction or reaction-regeneration cycles. Benefiting from moderate strength in the adsorption of reactants and intermediates, SACs showed outstanding activity and desired selectivity in alkane dehydrogenation. Significantly, SACs with a simple geometric structure and coordination environment can serve as a model to provide fundamental insights into the reaction mechanism at the atomic level. Therefore, the highly active Ir SACs are still sought after in alkane dehydrogenation.

In this work, we fabricated a series of highly dispersed Ir catalysts on nanodiamond@graphene (ND@G), including atomically dispersed Ir atom (Ir₁/ND@G), Ir sub-nanocluster (Irₙ/ND@G) and Ir NPs (IrNPs/

ND@G). Especially for the Ir₁/ND@G catalyst, the Ir SAs were stabilized by the Ir-C bond on the ND@G surface, resulting in a similar structure with supported single-site iridium pincer complexes. In n-butane dehydrogenation (BDH), Ir₁/ND@G showed a remarkable butane reaction rate (8.8 mol g$_{Ir}^{-1}$ h$^{-1}$) with high butene selectivity (95.6%). Even at a relatively low temperature (450 °C), Ir₁/ND@G showed a turnover frequency (TOF) of 0.48 s$^{-1}$, 19.2 times higher than Irₙ/ND@G and 34.3 times higher than IrNPs/ND@G. Significantly, we correlated the BDH activity with the average coordination numbers (CNs) of the Ir-Ir bond to reveal the structure-dependence from the nanoscale to the sub-nanoscale. We compared the Ir SAs with the catalysts of Pt SAs and Pd SAs to understand metal-dependence for BDH. Density functional theory (DFT) calculations results suggested that moderate adsorption of intermediates and easy desorption of butene on Ir SAs guaranteed high activity towards butene and remarkable catalyst stability.

## Results

### Preparation and characterization of highly dispersed Ir catalysts
ND@G, composed of the sp³ diamond core and highly defective sp² graphene outer shells, has been used as the support (Supplementary Fig. 1). The unique surface with abundant defects can trap and stabilize metal atoms by the metal−C bond. Its morphology and fine structure have been well studied in our previous reports[23–25]. A series of Ir/ND@G catalysts with different Ir loadings were prepared by the impregnation method or precipitation method, denoted as Ir₁/ND@G (0.02 wt%), Ir₁₊ₙ/ND@G (0.43 wt%), Irₙ/ND@G (1.3 wt%), and IrNPs/ND@G (1.5 wt%), respectively. The physicochemical parameters of all the Ir/ND@G catalysts were summarized in Supplementary Table 1. To characterize the atomic-scale structure of catalysts, aberration-corrected high-angle annular dark-field scanning transmission electron microscopy (HAADF-STEM) was employed. For Ir₁/ND@G, the Ir species was in the form of isolated Ir atoms without any Ir clusters and Ir NPs, as shown in Fig. 1a. The well-dispersed Ir atoms are highlighted by yellow circles in Fig. 1b−e. For Ir₁₊ₙ/ND@G, the Ir clusters of a few atoms (highlighted by pink circles) began to appear among abundant isolated Ir atoms (Fig. 1f−g and Supplementary Fig. 2). As the Ir loading increased to 1.3 wt%, the island-like Ir clusters (d = 0.77 ± 0.16 nm, marked in pink circles) with 10−13 atoms (without crystalline structure) were clearly observed for Irₙ/ND@G (Fig. 1h, i and Supplementary Fig. 3), indicating that the Ir species was well dispersed on the ND@G support. In contrast, in IrNPs/ND@G, the Ir NPs (nanoparticle diameter, d = 1.72 ± 0.30 nm) were located on the ND@G surface predominantly, together with a few isolated Ir atoms (Supplementary Fig. 4). The lattice spacing of Ir NPs was 0.22 nm, corresponding to the (111) facet of typical Ir NPs, implying the good crystallinity of the as-prepared Ir NPs on ND@G. Besides, from X-ray diffraction (XRD) profiles, the diffraction peaks were related to nanodiamond and graphite on the catalysts (Fig. 2a), indicating that the Ir species was highly dispersed on the ND@G surface, even for IrNPs/ND@G. These results are in good agreement with the HAADF-STEM observation.

In situ CO-diffuse reflectance infrared Fourier transform spectroscopy (CO-DRIFTS) is a powerful technology to study the surface structures of isolated Ir atoms, Ir clusters, and IrNPs in as-prepared catalysts. For Ir₁/ND@G, a pair of CO adsorption peaks were apparently observed at 2086 and 2028 cm$^{-1}$, respectively, which could be attributed to dicarbonyl species adsorbing on positively charged Ir species (Fig. 2b)[26–28]. The absence of other peaks suggested that no Ir multi-atomic species were present on ND@G. For Irₙ/ND@G and IrNPs/ND@G, except for peaks of dicarbonyl species, two peaks appeared at 2050−2030 and 1980−1960 cm$^{-1}$, respectively, which can be attributed to atop and bridged CO species adsorbing on metallic Ir species (Fig. 2c and Supplementary Fig. 5)[28,29]. Besides, for IrNPs/ND@G, a prominent peak of atop CO species demonstrated that large metallic Ir species have formed.

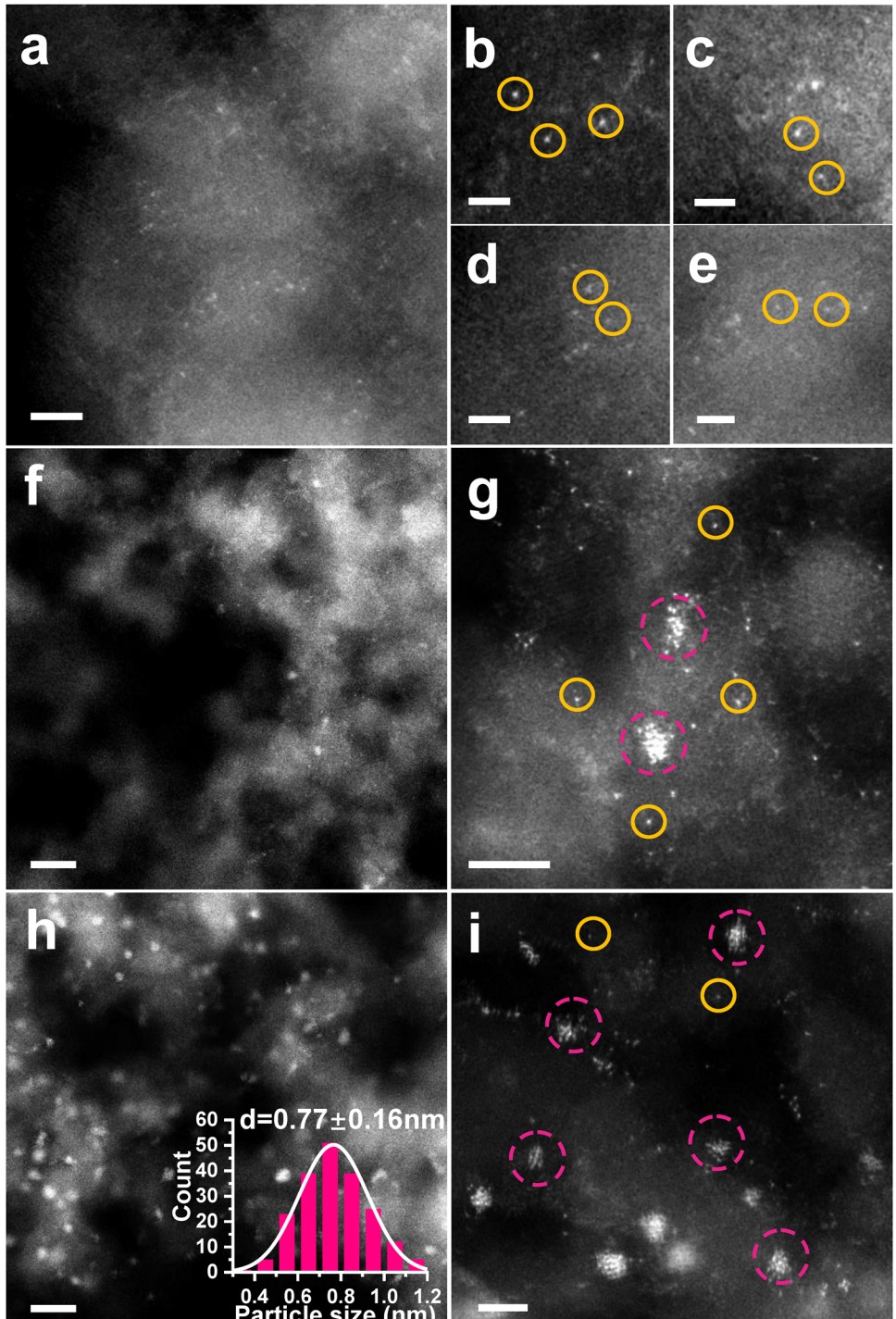

**Fig. 1 | Microscopic characterization of Ir$_1$/ND@G and Ir$_{1+n}$/ND@G and Ir$_n$/ND@G. a–e** HAADF-STEM images of Ir$_1$/ND@G. In the images, isolated Ir atoms are highlighted by yellow circles. **f, g** HAADF-STEM images of Ir$_{1+n}$/ND@G. In the images, Ir clusters are highlighted by the pink circles. **h, i** HAADF-STEM images of Ir$_n$/ND@G. Scale bars: **f, h**, 5 nm; **a, g, i**, 2 nm, and **b–e** 1 nm.

The Ir $L_3$-edge X-ray absorption near-edge structure (XANES) spectroscopy and X-ray photoelectron spectroscopy (XPS) measurements revealed the average oxidation states of Ir on these samples. Generally, as decreasing the size of supported metal particles, the fraction of metal−support interface accordingly increased. Metal particles with higher electron-deficient charge states emerge from the strong charge transfer between the metal particles and the support, which results in the formation of positively charged metal species[30,31]. For Ir$_1$/ND@G, the intensity of the white line was near

IrO$_2$, indicating the valence of Ir species was almost +4 valence[32,33], which can be resulted from the atomic dispersion (Supplementary Fig. 6). When the Ir loading increases, the Ir $4f_{7/2}$ peak shifted towards lower binding energy, indicating that Ir species was gradually close to the metallic state along with the formation of Ir clusters or NPs (Supplementary Fig. 7)[34,35]. To further study the fine structure and local environment of Ir species, Fourier-transformed (FT) $k^3$-weighted extended X-ray absorption fine structure (EXAFS) profiles were obtained (Fig. 2d). For Ir$_1$/ND@G, the Ir-C/O scattering at ~1.6 Å was

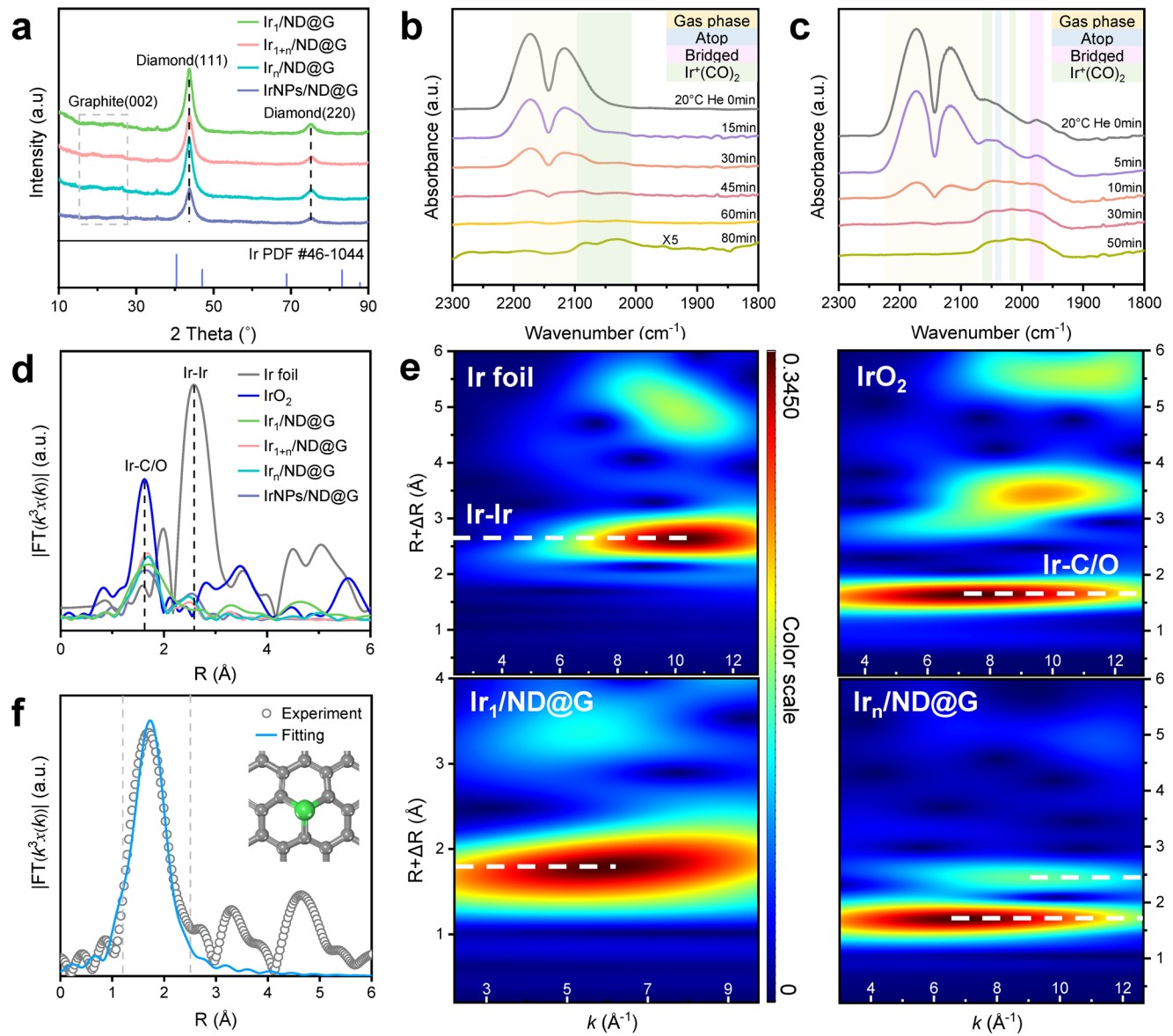

**Fig. 2 | Structure characterizations of as-prepared catalysts. a** XRD patterns of Ir$_1$/ND@G, Ir$_{1+n}$/ND@G, Ir$_n$/ND@G, and IrNPs/ND@G. **b** In situ CO-DRIFTS of Ir$_1$/ND@G. **c** In situ CO-DRIFTS of Ir$_n$/ND@G. **d** Fourier-transformed $k^3$-weighted EXAFS spectra of above as-prepared catalysts, Ir foil and IrO$_2$. **e** WT analysis of Ir$_1$/ND@G, Ir$_n$/ND@G, Ir foil, and IrO$_2$. **f** EXAFS fitting curve for Ir$_1$/ND@G and the optimized Ir-C$_3$ structure.

detected and no Ir-Ir scattering was observed, confirming that Ir species were atomically dispersed on ND@G. For Ir$_{1+n}$/ND@G, Ir$_n$/ND@G and IrNPs/ND@G, a distinct Ir-Ir scattering at ~2.6 Å was detected, and the average CN of Ir-Ir is 2.6, 3.5, and 3.6, respectively (Supplementary Table 2). Combined with the HAADF-STEM images, these results demonstrated the catalyst structure evolution from isolated Ir atoms to Ir clusters and then to Ir NPs, with the increase of Ir loading. The wavelet transformation (WT) of Ir $L_3$-edge EXAFS oscillations visibly displayed the different forms of Ir species in both the $k$ and R spaces. As shown in Fig. 2e, a maximum in the WT plot was observed at near 1.6 Å, which corresponds to the Ir-C/O backscattering in Ir$_1$/ND@G and Ir$_n$/ND@G. Moreover, another weak peak emerged at near 2.6 Å in Ir$_n$/ND@G, which was attributed to the Ir-Ir scattering, verifying the presence of Ir clusters with low CN. Quantitative chemical configuration analysis of Ir catalysts was carried out through the least-squared EXAFS fitting. The R space and $k$ space fitting results are shown in Fig. 2f and Supplementary Figs. 8, 9. The corresponding structure parameters are listed in Supplementary Table 2. Based on these results, the proposed local atomic structure

of Ir was constructed (see Fig. 2f). The isolated Ir atom was anchored over the defective sites of graphene by coordinating with three C atoms.

**Structure-dependence of Ir catalyst in alkane dehydrogenation**
To gain insight into the structure-dependence of catalysts for alkane dehydrogenation, the catalytic performance of different Ir catalysts was evaluated for BDH (Fig. 3a and Supplementary Table 3). For Ir$_n$/ND@G, the n-butane initial conversion was 19.2%, and the selectivity towards butene was 93.9% (Supplementary Fig. 10). The n-butane conversion dropped to 12.5%, and the value of $k_d$ (deactivation rate constant) was 0.0749 h$^{-1}$ in 10 h test (Fig. 3b). The stable structure on used Ir$_n$/ND@G suggested the rapid deactivation resulting from coke formation to block Ir active sites (Supplementary Fig. 11 and Supplementary Table 4). When the Ir loading decreased, the reaction rate and butene selectivity increased simultaneously (Fig. 3a). For Ir$_{1+n}$/ND@G, the reaction rate of n-butane (1.6 mol·g$_{Ir}$$^{-1}$·h$^{-1}$) was 2.7 times higher than that of Ir$_n$/ND@G (0.59 mol·g$_{Ir}$$^{-1}$·h$^{-1}$) (Fig. 3a). Moreover, $k_d$ decreased to 0.0405 h$^{-1}$ (Fig. 3b). The enhanced activity and stability indicated

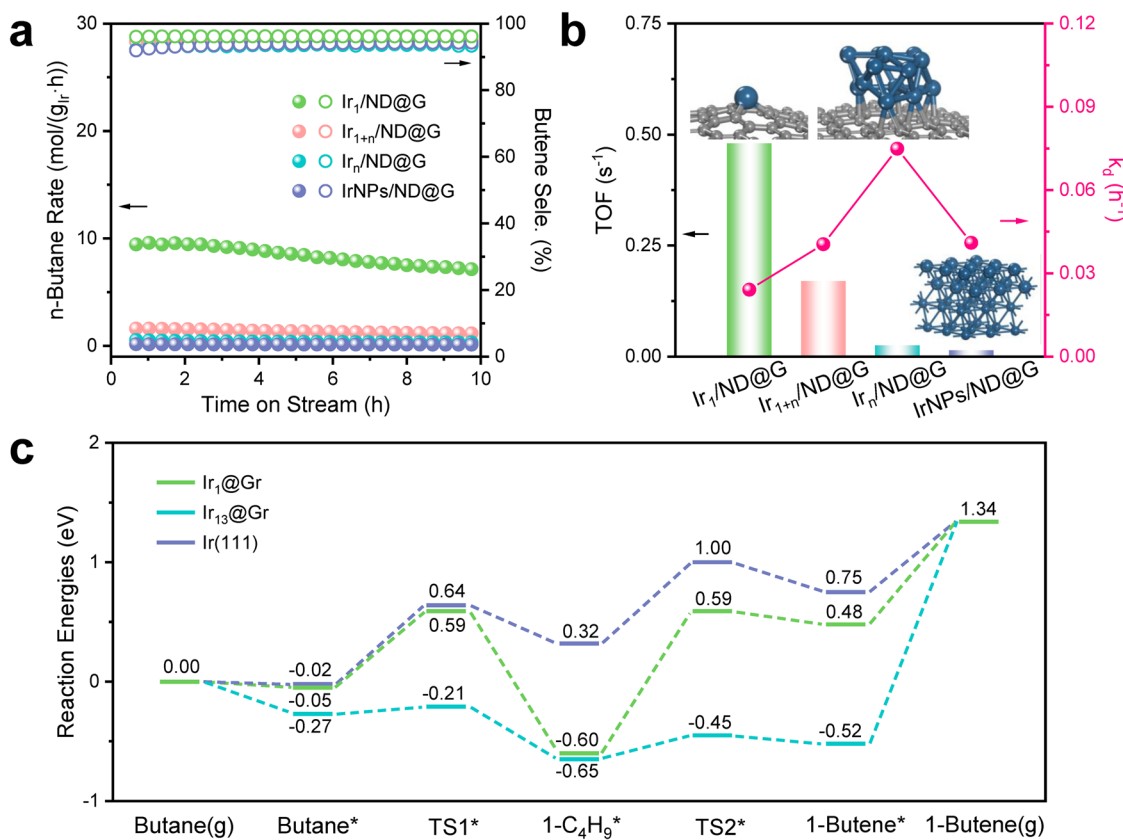

**Fig. 3 | Catalytic performance for BDH. a** n-Butane reaction rate and butene selectivity of $Ir_1/ND@G$, $Ir_{1+n}/ND@G$, $Ir_n/ND@G$, and IrNPs/ND@G. **b** The value of TOF and $k_d$. **c** Energy profiles of BDH on $Ir_1@Gr$, $Ir_{13}@Gr$, and Ir(111).

that the Ir-Ir CN might be the key factor for BDH. The structure of used $Ir_{1+n}/ND@G$ has been shown in Supplementary Fig. 11. Significantly, $Ir_1/ND@G$ with only 0.02 wt% Ir loading showed the highest activity and selectivity to butene (Fig. 3a). The reaction rate of n-butane on $Ir_1/ND@G$ reached 8.8 mol·$g_{Ir}^{-1}$·$h^{-1}$ and remained 7.1 mol·$g_{Ir}^{-1}$·$h^{-1}$ after 10 h, which was 5.5 times higher than that of $Ir_{1+n}/ND@G$ and 14.9 times higher than that of $Ir_n/ND@G$. Moreover, the butene selectivity was as high as 95% throughout the process. To compare the butene selectivity at higher butane conversion, the mass of the $Ir_1/ND@G$ catalyst has been increased (Supplementary Fig. 12 and Supplementary Table 5). The $Ir_1/ND@G$ showed the highest butene selectivity, suggesting the undesired side reaction was well-restrained in the absence of Ir-Ir bonds. On the $Ir_1/ND@G$, the n-butane initial conversion increased proportionally, confirming all the Ir atoms could directly participate in the reaction processes, including the adsorption and transformation of reactants. Moreover, the catalyst stability test was carried out for $Ir_1/ND@G$ by a 20-h and 50-h run (Supplementary Figs. 13, 14). The n-butane reaction rate remained constant at 6.0 mol·$g_{Ir}^{-1}$·$h^{-1}$ after 20 h. In long-term stability test for 50 h, the initial conversion of n-butane was 6.8% and still exhibited a considerable conversion of 4.1% after the 50 h reaction. The selectivity toward butene was as high as 95.7% after a 50 h reaction, and the value for $k_d$ was only 0.0107 $h^{-1}$. In contrast, IrNPs/ND@G, with few active Ir clusters and Ir SAs, showed delayed catalytic activity and selectivity (Supplementary Fig. 10).

Figure 3b illustrates the correlation between catalytic activity (or stability) and Ir species structure. The TOF increased with the structural evolution of Ir species from the nanoscale to the sub-nanoscale. The $Ir_1/ND@G$ with maximized atomic utilization showed a much higher n-butane reaction rate and TOF compared with the other catalysts with Ir ensemble sites, indicating that isolated Ir atoms were adequate for adsorbing n-butane and activating C-H bonds. Coke formation is a non-negligible cause for catalyst deactivation during BDH.

Notably, the reduced stability (increased $k_d$ value) was accompanied by the increasing size of Ir species. On $Ir_n/ND@G$, the $k_d$ value reached a maximum, indicating the atomically dispersed structure can suppress coke formation and retain outstanding catalytic activity during BDH. The amount of coke deposition on each used catalyst was measured by the thermogravimetric analysis (TGA) techniques, which was related to the size of Ir species on supported Ir catalysts (Supplementary Table 4). Additionally, the peak intensity ratio of D1 and G bands ($I_{D1}/I_G$) from Raman spectra is an important index to reflect the degree of graphitization on carbon materials. From the comparison of $I_{D1}/I_G$ between the fresh and used catalysts, lowered $I_{D1}/I_G$ on used catalysts suggests that increasing graphitic coke deposited during dehydrogenation (Supplementary Fig. 15)[17,36,37]. Besides, the Ir/ND@G samples were characterized after the BDH tests by STEM and XRD. After 10-h BDH, no sintering of Ir clusters or isolated Ir atoms into larger NPs was observed in any of the samples (Supplementary Figs. 11, 16). After 0.5 h BDH, Ir species of $Ir_1/ND@G$ was atomically dispersed, indicating isolated Ir sites were the real active sites during the reaction process (Supplementary Fig. 17). The STEM images of $Ir_1/ND@G$ after 10 and 20 h BDH showed that few isolated Ir atoms was aggregated into small clusters, which attenuated the remarkable activity of isolated Ir atoms (Supplementary Figs. 16, 18). As a result, structural evolution from SAs to clusters was a crucial factor for Ir sub-nanoscale catalytic deactivation. In addition, the reusability of $Ir_1/ND@G$ catalyst has been also examined (Supplementary Fig. 19). The n-butane conversion and butene selectivity have been almost restored in two regeneration cycles. The $k_d$ values of the two regeneration cycles were 0.0228 $h^{-1}$ and 0.0216 $h^{-1}$, respectively. The butene selectivity was above 94% throughout the reusability test. In consideration of the stability of $Ir_1/ND@G$ at the initial reaction step, in situ CO-DRIFTS has been used to accurately identify the structure of $Ir_1/ND@G$ under reaction conditions and reduction conditions. For $Ir_1/ND@G$ under

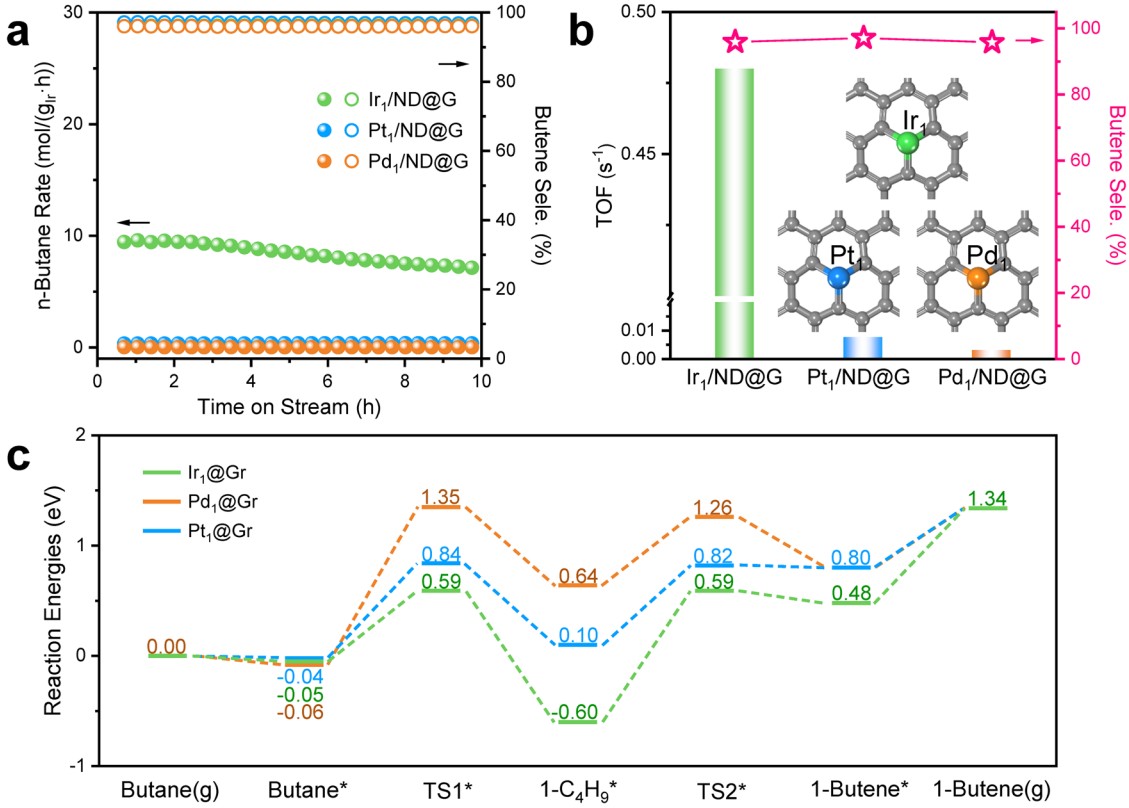

**Fig. 4 | Reaction performance for BDH. a** n-Butane reaction rate and butene selectivity of Ir$_1$/ND@G, Pt$_1$/ND@G, and Pd$_1$/ND@G. **b** The value of TOF and butene selectivity. **c** Energy profiles of BDH on Ir$_1$@Gr, Pt$_1$@Gr, and Pd$_1$@Gr.

reaction condition or reduction treatment, a pair of CO adsorption peaks were apparently observed at 2100–2086 cm$^{-1}$ and 2047–2017 cm$^{-1}$, respectively, which could be attributed to dicarbonyl species adsorbing on positively charged Ir species (Supplementary Figs. 20, 21). These results are consistent with those on the fresh sample (Fig. 2b). No peaks of atop and bridged CO species on Ir$^0$ multiatomic species have been observed, demonstrating isolated Ir atom bonded with C atoms was stable and reserved its positive valence at the initial period of the reaction. Therefore, we concluded that the Ir$_1$/ND@G catalyst exhibited decent activity, high C$_4$ olefins (or butene) selectivity and stability at relatively low temperatures, compared with the previously reported supported metal catalysts as displayed in Supplementary Table 6.

As shown in Fig. 3c, BDH into butene on model Ir$_1$@Gr, Ir$_{13}$@Gr, or Ir(111) surface (structural details in Supplementary Fig. 22) was computed by DFT to understand the activity difference induced by catalytic structure. Physical adsorption of butane was found on Ir(111) and Ir$_1$@Gr, which was indicated by their adsorption energies (−0.02 and −0.05 eV, respectively). But the stronger adsorption of butane occurred on Ir$_{13}$@Gr with the adsorption energies of −0.27 eV. Ir$_{13}$@Gr exhibits much higher dehydrogenation activity than Ir(111) and Ir$_1$@Gr, because of the strong adsorption of butane* (−0.27 eV), extremely low stepwise barrier (0.06 and 0.20 eV), and overall exothermicity (−0.52 eV) until the formation of 1-butene*. Different from Ir$_{13}$@Gr, butane dehydrogenation into 1-butene* is endothermic by 0.48 eV on Ir$_1$@Gr with an apparent barrier of 0.59 eV and is endothermic by 0.75 eV on Ir(111) with an apparent barrier of 1.00 eV. Therefore, the order of dehydrogenation activity is Ir$_{13}$@Gr > Ir$_1$@Gr > Ir(111). However, the difficult desorption of 1-butene* (1.86 eV) and overly superior dehydrogenation ability on Ir$_{13}$@Gr expanded the possibility of overdehydrogenation of 1-butene* and coke formation. Corrected Gibbs free energy profiles are shown in Supplementary Fig. 23. As a result, Ir$_1$@Gr showed high catalytic performance with moderate catalytic

activity and intermediate stability. Moreover, it should be noticed that the rate-determining step is significantly dependent on Ir species structure. On Ir$_1$@Gr and Ir(111), the steps of C-H activation play primary roles, determining the dehydrogenation activity during the overall process. On Ir$_{13}$@Gr, the strong adsorption of reaction intermediates makes C-H activation kinetically easy and further shifts the rate-determining step from C-H activation to butene desorption.

## Metal-dependence of noble metal catalysts in alkane dehydrogenation

Highly efficient Pt and Pd catalysts have been exploited for dehydrogenation reactions (such as alkane dehydrogenation and cycloalkane dehydrogenation) according to the higher activity towards C−H bond cleavage against C−C bond cleavage[25,38–42]. To disclose the cause for the spectacular activity over Ir$_1$/ND@G, Pt$_1$/ND@G, and Pd$_1$/ND@G were prepared by the impregnation method, and the nature of metal-dependence was investigated for the BDH reaction. For Pt$_1$/ND@G and Pd$_1$/ND@G, the structure and morphology had been described in our previous reports[23,25,43]. All the Pd atoms and Pt atoms were atomically dispersed on ND@G support without visible NPs or clusters.

As shown in Fig. 4, Supplementary Fig. 24, and Table 3, the results of catalytic performance over Ir$_1$/ND@G, Pt$_1$/ND@G, and Pd$_1$/ND@G were summarized. The Ir$_1$/ND@G containing only 0.02 wt% Ir showed a higher conversion of 4.6% than Pt$_1$/ND@G (1.2%) (Supplementary Fig. 24). Moreover, the reaction rate of n-butane on Ir$_1$/ND@G was 26 times higher than that of Pt$_1$/ND@G (0.34 mol·g$_{Ir}^{-1}$·h$^{-1}$) (Fig. 4a). Unexpectedly, the reaction rate of n-butane on Pd$_1$/ND@G was 0.0489 mol·g$_{Ir}^{-1}$·h$^{-1}$ (Fig. 4a), suggesting its relative inertness in C-H activation. The intrinsic activity of the above three samples was evaluated by normalizing the activity to metal atoms exposed on the surface and obtaining TOF (Fig. 4b). On Ir$_1$/ND@G, the TOF reached 0.48 s$^{-1}$, which was 62 times and 155 times higher than that of

Pt₁/ND@G and Pd₁/ND@G respectively. To reveal the nature of the different metals' dehydrogenation activity of n-butane, temperature-programmed surface reactions (TPSR) with the mixture of n-butane and D₂ were also conducted on the above three samples. Here, the combination of H or D atoms to generate H₂ or HD is assumed as a facile step in BDH. And the generation of HD indicates the first step of C−H activation. On Ir₁/ND@G catalyst, the appearance of HD was observed at 536 K (Supplementary Fig. 25), which was more favorable than that on Pt₁/ND@G (645 K) and Pd₁/ND@G (670 K). The results suggest that activation of the first C-H bond on Ir₁/ND@G is easier than that on Pt₁/ND@G and Pd₁/ND@G. Moreover, the onset temperature of butene formation was in the order of Ir₁/ND@G (669 K) < Pt₁/ND@G (745 K) < Pd₁/ND@G (763 K), suggesting that cleavage of the second C-H bond also influence the dehydrogenation activity. Combined with the above observation, C-H activation is a rate-determining step on SACs. Both the first and second steps of C-H activation can limit dehydrogenation activity in the whole BDH process. Moreover, C-H activation is significantly dependent on the metal species. BDH activity is in the order of Ir₁/ND@G > Pt₁/ND@G > Pd₁/ND@G.

To gain further mechanistic insight into metal-dependent BDH activities, DFT calculations were performed on Ir₁@Gr, Pt₁@Gr, and Pd₁@Gr (Supplementary Figs. 22, 26). As shown in Fig. 4c, the BDH into butene has a determining step of butane* → 1-C₄H₉* on Pt₁@Gr and Pd₁@Gr, which shows a barrier of 1.41 and 0.88 eV, respectively. On Ir₁@Gr, there is a determining step of 1-C₄H₉* → 1-butene* with a barrier of 1.19 eV. However, on Pt₁@Gr, the dehydrogenation of 1-C₄H₉* into 1-butene* has a barrier of 0.82 eV and is endothermic by 0.80 eV, indicating that the formation of 1-butene* is unstable and easy to hydrogenate into 1-C₄H₉*. Compared with Pt₁@Gr and Pd₁@Gr, butane* dehydrogenation into 1-C₄H₉* has a lower barrier (0.64 vs. 0.88 and 1.41 eV) and more exothermicity (−0.55 vs. 0.14 and 0.70 eV) on Ir₁@Gr. Thus, the formation of 1-C₄H₉* was favored thermodynamically and kinetically. An accumulation in the content of 1-C₄H₉* on Ir₁@Gr can be predicted and will increase the possible formation of 1-butene*. In addition, the calculated d-band centers for Ir₁@Gr, Pt₁@Gr, and Pd₁@Gr are −2.64, −4.09, and −4.13 eV, respectively (Fig. 5). This result suggests stronger adsorption will occur on Ir₁@Gr than Pt₁@Gr and Pd₁@Gr, which is consistent with Fig. 4c. Based on the experimental observations and theoretical calculations, the adsorption of intermediates is depended on the metal species. The stronger adsorption of intermediates is the key to the higher activity on Ir SAs than that of Pt SAs or Pd SAs.

On Pt₁/ND@G, the poorer activity for dehydrogenation steps was influenced by the weaker adsorption of intermediates. We propose that a higher activity on Pt would require more unsaturated Pt sites. The triangular Pt₃ cluster (Pt₃/ND@G) reported in our previous work was an efficient catalyst in BDH[43]. The n-butane reaction rate on Pt₃/ND@G was nearly six times higher than that on Pt₁/ND@G. The reason for enhanced activity would attribute to the adsorption sites with more unsaturation. The calculated d-band center for Pt₃@Gr was −2.14 eV, indicating that stronger adsorption will occur on Pt₃@Gr in contrast with Pt₁@Gr (−4.09 eV) (Fig. 5). Moreover, the d-band center for Pt₃@Gr was close to Ir₁@Gr (−2.64 eV) and located near Fermi level, indicating the stronger adsorption of intermediates and products. The TOF of Pt₃/ND@G (0.12 h⁻¹) was four times lower than Ir₁/ND@G (0.48 h⁻¹), indicating that the rate-determining step on Pt₃@Gr would shift from C-H activation towards butene desorption due to the stronger adsorption of products. According to the above proposal, we would further improve the catalytic activity on Ir₁ND@G, if constructing one cluster with more unsaturated sites. Unexpectedly, the calculated d-band center for Ir₁₃@Gr was −1.45 eV and closer to the Fermi level, compared with Ir₁@Gr and Pt₃@Gr (Fig. 5). Owing to the strong adsorption behavior, rate-determining step on Ir₁₃@Gr located at the step of butene desorption. The difficult desorption of butene led to poor catalytic performance in BDH (Fig. 3c).

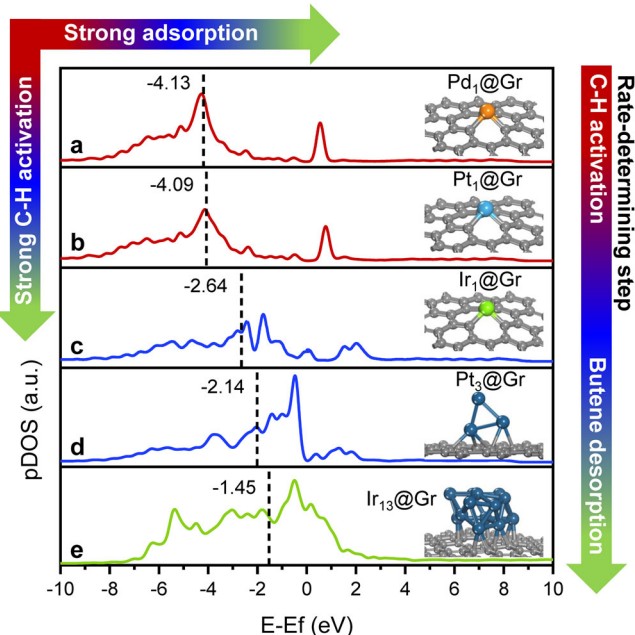

**Fig. 5 | DOS curves for d orbitals of Pd, Pt, and Ir atom.** The d-band center was highlighted by a dotted line. **a−e** DOS curve was for Pd₁@Gr, Pt₁@Gr, Ir₁@Gr, Pt₃@Gr, and Ir₁₃@Gr, respectively. The *d*-band center was −4.13, −4.09, −2.64, −2.14, and −1.45 eV, respectively. The structure of these models was also presented. The arrow in the upper left represents the trend of adsorption strength and C-H activation. The arrow in the right represents the trend of shifted rate-determining step.

Furthermore, temperature-programmed desorption of butene (C₄H₈-TPD) and TGA have been carried out to solidify the above conclusion from DFT calculations (Supplementary Fig. 27 and Table 4). Therefore, the highly efficient catalyst would not be determined by a more unsaturated structure but by moderate strength of adsorption, strongly dependent on the metal species. The Ir₁ND@G with moderate adsorption behavior showed the highest catalytic performance with moderate activity and stability. Besides BDH, the nature of structure-dependence and metal-dependence have been expanded to PDH. A similar discipline has been found. For the nature of structure-dependence, Ir₁ND@G showed higher activity than Irₙ/ND@G and IrNPs/ND@G (Supplementary Fig. 28). For the nature of metal-dependence, Ir₁ND@G showed high activity, and Pt₁/ND@G and Pd₁/ND@G was almost inactive for PDH (Supplementary Fig. 29).

## Discussion

In summary, a series of highly dispersed Ir catalysts were successfully synthesized on ND@G support for BDH, inspired by single-site pincer-ligated iridium complexes in homogeneous catalysis. To gain insights into the nature of structure-dependence and metal-dependence of catalyst, we correlated the BDH activity with different Ir structures ranging from nanoscale to sub-nanoscale and also compared the Ir SAs catalyst with Pt SAs and Pd SAs. Compared to Ir sub-nanoclusters and NPs, Ir SAs showed the highest n-butane conversion and the best butene selectivity (against both side reactions and coke formation). The suitable strength of intermediates adsorption shifted the rate-determining step from butene desorption to C-H activation. And thus, moderate dehydrogenation activity and easy desorption of 1-butene ensure the stability and outstanding catalytic performance of Ir SAs. Moreover, the adsorption of intermediates is also determined by the metal species, which is the main reason for the higher activity on Ir SAs than that of Pt SAs or Pd SAs. Our results illustrate that the isolated metal sites, not just the ensemble sites, can be adaptable toward both reactant adsorption/activation and product desorption. The good dehydrogenation

capacity and moderate adsorption behavior are the key to the high catalytic activity and butene selectivity, which are strongly dependent on metal structure and content. Importantly, such nature of structure-dependence and metal-dependence pave a new path to design catalysts with high activity, selectivity, and stability for efficient dehydrogenation.

## Methods

### Materials
Nanodiamond (ND) powders were purchased from Beijing Grish Hitech Co., Ltd, China. Analytical grade Chloroiridic acid ($H_2IrCl_6 \cdot 6H_2O$) as metal precursors were purchased from Aladin Chemical Reagent Inc. Sodium formate (HCOONa) and ammonium hydroxide aqueous solution ($NH_4OH$, 25–28%) were purchased from Sinopharm Chemical Reagent Co. Ltd. China).

### Preparation of ND@G
Nanodiamond powder was annealed at 1100 °C under Ar flow (100 mL/min) for 4 h and then cooled to room temperature. The obtained ND@G powder was further purified to remove impurities by concentrated hydrochloric acid for 24 h, and then washed with deionized water. Finally, the purified ND@G was dried in a vacuum at 80 °C for 48 h.

### Preparation of $Ir_1$/ND@G and IrNPs/ND@G
The catalysts were prepared by the impregnation method. Typically, a certain amount of $H_2IrCl_6 \cdot 6H_2O$ solution (10 g/L) was added into 2 mL ethanol, and the mixture was ultrasonically treated for 5 min. Then, 200 mg purified ND@G powder was added to the above solution, and the mixture was ultrasonically treated for 5 min to obtain a homogenous suspension. The mixture was stirred at room temperature for 24 h and then dried at 60 °C for another 12 h. Finally, the $Ir_1$/ND@G sample was reduced in $H_2$ (10 vol% $H_2$ in $N_2$, flow rate = 30 mL/min) at 200 °C for 1 h, and the IrNPs/ND@G sample was reduced in $H_2$ (10 vol% in $N_2$, flow rate = 30 mL/min) at 450 °C for 2 h. Limited by the wet chemistry synthesis method, the maximum Ir loading of SA is 0.1%, confirmed by ICP-OES. Benefiting from the low loading amount, many high-quality works reported important results in fundamental research[44–48]. Herein, we chose the $Ir_1$/ND@G with 0.02 wt% Ir as the major sample.

### Preparation of $Ir_{1+n}$/ND@G and $Ir_n$/ND@G
The catalysts were prepared by the deposition-precipitation method. Typically, 200 mg purified ND@G was dispersed into 25 mL deionized water in a 100 mL round-bottom flask, and the mixture was ultrasonically treated for 30 mi. Then, the pH value of the above suspension was adjusted to about 10 by the addition of HCOONa. Afterward, a certain amount of $H_2IrCl_6 \cdot 6H_2O$ solution (10 g/L) was introduced into the above suspension, and then adjusted the pH value to 7. The mixture was stirred at 100 °C in an oil bath for 1 h. And then, the obtained mixture was cooled to room temperature, washed with deionized water, and dried in a vacuum at 60 °C for 12 h. Finally, the solid sample was reduced in $H_2$ (10 vol% $H_2$ in $N_2$, flow rate = 30 mL/min) at 450 °C for 2 h.

### Catalytic performance
The catalytic performance for the BDH was conducted in a fixed-bed stainless steel micro-reactor with a quartz lining under atmosphere pressure at 450 °C with 20 mg catalysts. A gas mixture of 2% $H_2$ and 2% n-$C_4H_{10}$ with He balance (flow rate = 15 mL min$^{-1}$, GHSV = 45,000 mL g$^{-1}$ h$^{-1}$) was introduced. The effluent mixture gas was analyzed by online gas chromatography (Agilent 7890 with an FID and a TCD detector).

The n-butane conversion, $C_4$ olefin (butene and 1, 3-butadiene) selectivity, butene selectivity, and n-butane reaction rate were calculated using the following equations:

$$\text{n-Butane conversion} = \frac{\text{Mole of the reacted n-butane}}{\text{Mole of inlet n-butane}} \times 100\% \quad (1)$$

$$C_4 \text{ olefin Selectivity} = \frac{\text{Mole of(butene formed} + 1, 3\text{-butadiene formed)}}{\text{mol of reacted}} \times 100\% \quad (2)$$

$$\text{Butene selectivity} = \frac{\text{Mole of (butene formed)}}{\text{mol of reacted}} \times 100\% \quad (3)$$

$$\text{n-Butane reaction rate} = \frac{\text{Flow rate of n-butane} \times \text{conversion} \times 60}{\text{Ir weight} \times 22.4} \quad (4)$$

The n-butane conversion adapted to calculate turnover frequency (TOF) was below 15%. TOF of the catalysts was calculated using the following equation:

$$\text{TOF} = \frac{\text{Mole of n-butane conved per second}}{\text{Mole of active metal} \times \text{dispersion}} \quad (5)$$

The catalyst stability was described by a first-order deactivation model:

$$k_d = \frac{\ln\left(\frac{1-C_f}{C_f}\right) - \ln\left(\frac{1-C_i}{C_i}\right)}{t} \quad (6)$$

where $C_i$ is the initial conversion after reaction 0.33 h; $C_f$ is the final conversion after reaction 9.75 h; $t$ represents the reaction time (h); and $k_d$ is the deactivation rate constant (h$^{-1}$) that is used to evaluate the catalyst stability (the higher $k_d$ value is, the lower the stability).

The catalytic performance for the PDH was conducted in a fixed-bed stainless steel micro-reactor with a quartz lining under atmosphere pressure at 500 °C with 50 mg catalysts. A gas mixture of 5% $C_3H_8$ with He balance (flow rate = 15 mL min$^{-1}$, GHSV = 18,000 mL g$^{-1}$ h$^{-1}$) was introduced. The effluent mixture gas was analyzed by online gas chromatography (Agilent 7890 with an FID and a TCD detector).

The propane conversion, propene selectivity, and TOF were calculated using the following equations:

$$\text{Propane conversion} = \frac{\text{Mole of the reacted propane}}{\text{Mole of inlet propane}} \times 100\% \quad (7)$$

$$\text{Propene selectivity} = \frac{\text{Mole of (propene formed)}}{\text{mol of reacted}} \times 100\% \quad (8)$$

The propane conversion adapted to calculate TOF was below 10%. TOF of the catalysts was calculated using the following equation:

$$\text{TOF} = \frac{\text{Mole of propane conved per second}}{\text{Mole of active metal} \times \text{dispersion}} \quad (9)$$

### Regeneration tests
After the initial run, the catalyst was calcinated in 20% $O_2$ (30 mL/min) at 300 °C for 1.5 h and reduced in pure $H_2$ at 500 °C for 4 h to remove O sites. Finally, a gas mixture of 2% $H_2$ and 2% n-$C_4H_{10}$ with He balance (flow rate = 15 mL min$^{-1}$, GHSV = 30,000 mL g$^{-1}$ h$^{-1}$) was introduced, regarded as one regeneration cycle.

## Catalyst characterization

HRTEM images were taken by an FEI Tecnai G2 F20 working at 200 kV. HAADF-STEM images were recorded by a JEOL JEM ARM 200CF aberration-corrected cold field-emission scanning transmission electron microscope at 200 kV. The sample powder has been dispersed into ethanol and ultrasonically treated for 5 min to obtain a homogenous suspension. And then, we extracted the supernatant of 10 μL and dropped it onto holey carbon-coated copper grids in the atmosphere. XPS were carried out at ESCALAB 250 instrument with Al Kα X-rays (1489.6 eV, 150 W, 50.0 eV pass energy) and the C 1$s$ peak at 284.6 eV as internal standard. Before XPS characterization, the samples were reduced by 10% $H_2$ and collected in the atmosphere. XRD patterns were obtained by using a D/MAX-2500 PC X-ray diffractometer with monochromatized CuKα radiation (λ = 1.54 Å). Before XRD characterization, the samples were reduced by 10% $H_2$ and collected in the atmosphere. In situ CO-DRIFTS over different Ir catalysts were recorded on a Thermo Scientific Nicolet IS10 Fourier transform infrared spectrometer equipped with a high-temperature and high-pressure chamber and an MCT detector. Before measurement, the unreduced samples were diluted with KBr were placed in the sample holder, and reduced in $H_2$ (5 mL/min) for 1 h at the desired reduction temperature. And then, the sample holder was cooled to 293 K under He atmosphere. CO adsorption was carried out at room temperature in a flow of 5%CO/He (5 mL/min). Until the surface is completely covered by CO, the gas was switched to He (5 mL/min), and the spectrum was continuously recorded at 293 K. For in situ CO-DRIFTS of Ir$_1$/ND@G under reaction condition, the samples with KBr were placed in the sealed chamber and reduced in 10% $H_2$ (5 mL/min) for 1 h at 200 °C. After the chamber was heated to 450 °C in He, the gas was switched to reaction gas (2% $C_4H_{10}$, 2% $H_2$, He balance) and treated for 1 h at the reaction condition. And then, the sealed chamber was cooled to 20 °C in reaction gas. CO adsorption was carried out at 20 °C in a flow of 5% CO/He (5 mL/min). Until the surface is completely covered by CO, the gas was switched to He (5 mL/min) and the spectrum was continuously recorded at 20 °C. Furthermore, in order to evaluate the structure of the Ir$_1$/ND@G catalyst even in the $H_2$ atmosphere under the reaction temperature, the reduction temperature was raised to 450 °C (reaction temperature). For in situ CO-DRIFTS of Ir$_1$/ND@G under reduction treatment, the samples with KBr were placed in the sealed chamber and reduced in 10% $H_2$ (5 mL/min) for 1 h at 200 °C. The chamber was heated to 450 °C in He after reduction, the gas was switched to 10% $H_2$ and treated for 1 h at 450 °C. And then, the sealed chamber was cooled to 20 °C. CO adsorption was carried out at 20 °C in a flow of 5% CO/He (5 mL/min). Until the surface is completely covered by CO, the gas was switched to He (5 mL/min) and the spectrum was continuously recorded at 20 °C. The Brunauer−Emmett−Teller (BET) surface area, BJH pore volume, and average pore diameter of the as-prepared samples were measured by $N_2$ physisorption on the Micrometrics ASAP-2020 instrument. Before BET characterization, the samples were reduced by 10% $H_2$ and collected in the atmosphere. Elemental analysis of Iridium in the solid catalysts was detected by ICP-OES analysis. Before ICP-OES characterization, the samples were reduced by 10% $H_2$ and collected in the atmosphere. XAFS measurement at Ir $L_3$-edge (11,215 eV) was measured at the beamline BL14W1 station in Shanghai Synchrotron Radiation Facility (SSRF). The focused beam was tuned by the Si (111) double-crystal monochromators. Ir foil and IrO$_2$ were used as standards. The samples were measured in fluorescence mode, using a Lytle detector to collect the data. For the quasi-in situ XAS spectra of reduced catalysts, the samples were reduced in a fixed-bed reactor with 10 vol% $H_2/N_2$ at corresponding temperatures. After it was cooled to atmospheric temperature, the reactor was sealed with two globe valves and transferred to the glove box without exposure to air. And then, the prepared catalysts were sealed in a measurement plate by Kapton films under Ar protection.

The whole process was performed in the glove box. The reduced sample could not be oxidized by this sample preparation method for XAS data collection. For Ir$_1$/ND@G, a 32-channel solid detector was to achieve high data quality. All XAFS spectra were processed and analyzed by the Ifeffit package. The amount of carbon deposition was measured through the combustion of spent Ir/ND@G catalysts on the model STA 409 PC/PG thermogravimetric analyzer (NETZSCH). After BDH, the samples were collected in the air. UV-Raman spectroscopy was performed on powder samples by using the HORIBA LabRam HR Raman spectrometer, and the excitation wavelength was 325 nm with a power of 0.2 mW (exposure 20 s, accumulated four times). Before UV-Raman characterization, the samples before or after BDH were prepared in the air. In situ TEM is a powerful tool to verify the real structure of active species under a reaction atmosphere. However, due to the dramatically increased electron scattering from gas-cell membranes (usually two 50 nm-thick SiN films both above and beneath the 5–6-nm-thick nanodiamond sample) and from the $H_2$ atmosphere, as well as the thermal turbulence in 450 °C high temperature, the originally weak signal of single Ir atoms cannot be detected, considering TEM capacities. For in situ XAFS, the low loading amount (0.02 wt%) and the harsh reaction condition (450 °C) also lead to a noisy and weak signal of single Ir atoms. It is difficult to detect and analyze the precise structure information. Therefore, current eTEM and in situ XAFS techniques cannot provide the solid but desired evidence on the structure of Ir$_1$/ND@G under reaction conditions. To verify the catalytic structure, the HAADF-STEM images of the Ir$_1$/ND@G after 0.5 h BDH at 450 °C have been provided (Supplementary Fig. 18).

## Temperature-programmed surface reaction (TPSR)

The TPSR experiments for the BDH were conducted in a fixed-bed reactor with a quartz lining under the atmosphere. About 10 mg SACs were set in a 4 mm diameter quartz tube and were reduced in $D_2$ (10% $D_2$/Ar, flow rate = 20 mL/min) at 473 K for 2 h. About 10 mg Ir$_n$ and IrNPs catalysts were reduced in $D_2$ (10% $D_2$/Ar, flow rate = 20 ml/min) at 723 K for 2 h. After the catalyst was cooled to room temperature, the gas was switched to 10% n-$C_4H_{10}$/Ar (flow rate = 20 mL/min) and 10% $D_2$/Ar (flow rate = 2 mL/min), and the purging was continued for two hours to stabilize the signal. Then TPSR was operated at the programmed rising temperature for 5 K/min. A mass detector (OMNI StarTM GSD 350) was used to analyze the signal of m/z = 2 ($H_2$), 3 (HD), 4 ($D_2$), 56 ($C_4H_8$), and 58 ($C_4H_{10}$).

## Temperature-programmed desorption (TPD)

Temperature-programmed desorption of butene ($C_4H_8$-TPD) were performed in a quartz-bed flow reactor with an online mass spectrometer (MS, Pfeiffer OMNIstarTM). Typically, 50 mg samples were reduced in $H_2$ (10 % $H_2$/He, flow rate = 20 mL/min) at 200 °C for 2 h and treated in He (20 mL/min) to remove surface H species until the catalyst was cooled to room temperature. Then, the gas was switched to a flowing 2% $C_4H_8$/He (20 mL/min) and treated at room temperature for 2 h. After that, the system was swept in a flowing He streams (20 min mL/min) until a stable baseline was obtained. The temperature of the catalyst was then increased from 20 to 420 °C at the programmed rising temperature for 5 °C/min.

## Computational details

All DFT calculations with Perdew−Burke−Ernzerhof (PBE) functional[49] were performed by Vienna ab initio simulation package (VASP)[50,51]. The projector augmented wave (PAW)[52,53] potential was used to describe the interaction between ion and electron. A cutoff energy is set by 500 eV, the convergence tolerance is $10^{-5}$ eV and 0.02 eV/Å for electronic and ionic optimizations, respectively. By removing a carbon atom, there forms a carbon defect on a graphene layer (5 × 5). Pd$_1$/ND@G, Ir$_1$/ND@G, and Pt$_1$/ND@G models are constructed by

placing metal atoms at a carbon defect of a graphene layer. On the basis of EXAFS data, a two-layer $Ir_{13}$ cluster is doped in the carbon defect to construct an $Ir_{13}$/ND@G model. A $p(3 \times 3)$ Ir(111) surface with four layers was modeled and the top layers were fully relaxed as well as the bottom layers were fixed. The vacuum layers are set by 20 Å. The Brillouin zone was sampled with a grid of $3 \times 3 \times 1$ Monkhorst Pack grid[54]. The Gaussian smearing method was used with a smearing width of 0.05 eV. Constrained scan[55] combined with a DIMER method[56] were used to search the transition states, which were confirmed with only one imaginary frequency by frequency calculations. Gibbs free energies at 723 K and 1 atm were corrected with the VASPKIT mode[57].

## Data availability

The data supporting this article and other findings are available from the corresponding authors upon request. Source data are provided with this paper.

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

## Acknowledgements

This work was supported by the National Key R&D Program of China (2022YFA1504500, 2022YFB4003100, 2021YFA1502802 and 2021YFA1501100), the National Natural Science Foundation of China (92145301, U21B2092, 21961160722, 91845201, and 22072162), Chinese Academy of Sciences (172GJHZ2022028MI), Shenyang Young Talents Program (RC210435), Dalian National Lab for Clean Energy (DNL Cooperation Fund 202001), and China Petroleum & Chemical Corporation (No. 420043-2). D.M. acknowledges support from the Tencent Foundation through the EXPLORER PRIZE and New Cornerstone Investigator Program. N.W. and X.Ca. acknowledge the funding support from the Research Grants Council of Hong Kong (Project Nos. N_HKUST624/19 and 16306818). The XAS experiments were conducted in Beijing Synchrotron Radiation Facility (BSRF) and Shanghai Synchrotron Radiation Facility (SSRF). The authors also thank Prof Shuai Wang for his kind support in the discussion of the reaction mechanism and Maolin Wang for his effort in X-ray absorption fine structure spectroscopic measurements.

## Author contributions

H.L. and D.M. conceived the research. X.Ch. conducted material synthesis and carried out the catalytic performance test. X.Q. and Z.J. conducted the X-ray absorption fine structure spectroscopic measurements and analyzed the data. Y.J., P.R., and X.W. performed the DFT calculations. X.Ca. and N.W. contributed to the aberration-corrected high-angle annular dark-field scanning transmission electron microscopy. P.M. and C.L. conducted the X-ray photoelectron spectroscopy measurements. J.D. performed some of the synthesis experiments. The manuscript was primarily written by X.Ch., D.X., H.L., and D.M. All authors contributed to discussions and manuscript review.

## Competing interests

The authors declare no competing interests.
