## [Peer Review File · Nature Communications]

Atomically Dispersed Ir Catalysts for Efficient n-Butane Dehydrogenation: Structure-dependence and Metal-dependenceREVIEWER COMMENTS

Reviewer #1 (Remarks to the Author):

This paper describes a unique Ir catalyst with isolated Ir ions bonded to nano-diamonds or nano-carbon, which has high selectivity and high turnover rate for butane dehydrogenation. The catalysts are characterized by STEM, XAS and DFT. The results are incomplete to precisely define the unique structure of this catalyst. In addition, the reaction selectivity, kinetics, life and regenerability are not addressed. While the selectivity is high, the selectivity at higher conversions is not given. The loading of the catalyst is 0.02% Ir, which is too low for practical applications. In addition, all Ir catalysts including Ir NPs appear to have similar high butene selectivity. While the TOR of these are lower, the Ir loadings are higher making it possible to obtain higher conversions. Finally, butane dehydrogenation catalysts will require regeneration at some time on stream. With C based supports, it is unlikely that these can be regenerated.

Specific comments:

1. The Ir¹ catalyst are characterized by IR, XPS, STEM and XAS. It is necessary to determine that these are single Ir sites and their coordination geometry with number of bonds and bond distances and oxidation state. In addition, these need to be determined under reaction or treatment conditions, which represent the catalyst during butane dehydrogenation. These are required for proper DFT modeling of the active phase.

a. Because of the very low 0.02% Ir loading the XPS data is inconclusive as to the oxidation state. The top spectrum in Figure 6 is mostly noise and unconvincing. As a result, the exact oxidation state cannot be determined. It is not clear if these were taken after reduction at butane dehydrogenation conditions, or in air. If the latter this result is not representative of the active catalysts.

b. The EXAFS of the Ir¹ catalyst is similarly noisy. The EXAFS at $k >$ than about 7 Å⁻¹ is unreliable, which makes fitting for the FT inaccurate, see figure S8a and Table S2.

c. The STEM shows isolated Ir, at least when exposed to air. Are catalysts after butane dehydrogenation still isolated sites. The samples should be run in an eTEM cell, so that they can be treated in 450C in H₂ and then analyzed without exposure to air?

d. Until the exact structure and oxidation state of the Ir¹ is determined, the results from the DFT are not reliable. What if these are very small metallic Ir clusters under reaction conditions? The structure of the isolated pre-catalyst is not representative of the catalytic properties.

The characterizations of the novel Ir¹ catalysts fail to define the active oxidation state and coordination geometry.

2. The catalytic performance is incomplete. All Ir catalysts appear to have high C₄ olefin selectivity of greater than about 98%. The selectivity is typically lower at higher conversions; thus, these need to be measured at different conversions and compared at the same selectivity. Commercial catalysts operate at 40% propane conversion (at 1 atm C₃). What is the selectivity of each catalyst at these conversions? For dehydrogenation catalysts cycle length and regenerability are key metrics. What is the deactivation rate (and cycle length) for these catalysts? Can these be regenerated? Can the C and/or diamond

support be regenerated?

3. Single site catalysts on graphite and N-doped graphite are well known in electrochemistry for metals like Co, Ru, Fe and others. The metal loadings of these are often > 1% and display unique catalytic properties compared to metallic NPs. The Ir1 catalyst has 0.02% Ir. Is this the maximum loading? Even if the TOR is higher than Ir NP, the rate/g will be low. If these can't be prepared at higher loading, it is unlikely that these will be used for any practical reaction.

Preparation of single site catalysts with different properties, which are different from metallic NPs is well known for several metals. These occur on defect sites on C and N-doped C, similar to those described here. The characterizations also need to be obtained under in situ conditions similar to butane dehydrogenation to confirm that the as synthesized structure is the same as the catalytic structure. The accuracy and precision of these in situ characterizations also needs significant improvement to define both the local coordination geometry (number of bonds and bond distance) and active oxidation state. Then the modeling needs to reflect these active site structures, which may be different from those currently modeled.

This paper is not recommended for publication and should be sent to a more specialized catalysis journal after additional characterizations are completed.

Reviewer #2 (Remarks to the Author):

The work by Ma et al. on alkane dehydrogenation is a novel study of metal and structure dependence of activity and selectivity for this reaction. Some points need to be clarified after publication in NC:

- The reaction rate almost decreases to half of the initial value after 10 h (Fig. 4) or 20h (Fig S11). Longer TOS values are needed to check for the deactivation of the active sites.
- Experimental characterization of the sample after such TOS is needed to confirm the robust, atomically dispersed, and isolated nature of the active site.
- Comparison with TON and TOF values from the literature, as well as selectivities and stability is required. What is the role of either nanodiamond, graphene, or noble metal site?
- Substrate scope with different alkanes is suggested to find trends between substrate structure and catalyst performance. The scope of graphene and nanodiamond with different oxidation degrees will also unravel the role of the support and its acid-base properties, as well as metal-support interactions.

Reviewer #3 (Remarks to the Author):

The authors present an interesting study of highly dispersed Ir, including atomically dispersed Ir. The as-prepared catalysts are carefully characterized, and catalytic testing shows promising butane dehydrogenation rates and relatively good catalyst stability.

Most of the text attributes the high activity of the Pt₁/ND@G catalyst to the atomic dispersion. However, Figure S10 shows that after reaction, there isn't any evidence that single atoms are still present on the catalyst. There is some mention that the catalyst is covered in coke, but this isn't elaborated and whether this could affect the STEM detection of individual Ir atoms.

Ultimately this casts doubt into whether the active catalyst actually functions as single atoms. It is clear that it starts out that way, but perhaps it is all converted to the small Ir clusters seen in Figure S10b. It is definitely clear that synthesizing the catalyst as isolated Ir atoms leads to smaller Ir particles over time during reaction, but since the Iridium loading was smaller during these reactions, it isn't fully clear if this is due to the atomic dispersion or the lower loading overall.

The catalyst synthesis and catalytic results are both of high quality and worthy of publication, but the claim that the active catalyst is an isolated atom requires some careful consideration before it can be made.

REVIEWER COMMENTS

Reviewer #1 (Remarks to the Author):

This paper describes a unique Ir catalyst with isolated Ir ions bonded to nano-diamonds or nano-carbon, which has high selectivity and high turnover rate for butane dehydrogenation. The catalysts are characterized by STEM, XAS and DFT. The results are incomplete to precisely define the unique structure of this catalyst. In addition, the reaction selectivity, kinetics, life and regenerability are not addressed. While the selectivity is high, the selectivity at higher conversions is not given. The loading of the catalyst is 0.02% Ir, which is too low for practical applications. In addition, all Ir catalysts including Ir NPs appear to have similar high butene selectivity. While the TOR of these are lower, the Ir loadings are higher making it possible to obtain higher conversions. Finally, butane dehydrogenation catalysts will require regeneration at some time on stream. With C based supports, it is unlikely that these can be regenerated.

Specific comments:

1. The Ir^I catalyst are characterized by IR, XPS, STEM and XAS. It is necessary to determine that these are single Ir sites and their coordination geometry with number of bonds and bond distances and oxidation state. In addition, these need to be determined under reaction or treatment conditions, which represent the catalyst during butane dehydrogenation. These are required for proper DFT modeling of the active phase.
 - a. Because of the very low 0.02% Ir loading the XPS data is inconclusive as to the oxidation state. The top spectrum in Figure 6 is mostly noise and unconvincing. As a result, the exact oxidation state cannot be determined. It is not clear if these were taken after reduction at butane dehydrogenation conditions, or in air. If the latter this result is not representative of the active catalysts.

Response: We appreciate the reviewer for the nice suggestion. The XPS data is indeed noisy owing to such low metal loading of the Ir₁/ND@G catalyst (i.e., 0.02%). Following the suggestion, we replaced the XPS data with the XANES spectra of better quality in the revised supporting information (SI) (**Supplementary Figure 6**). The white line intensity of fresh Ir₁/ND@G was similar to that of IrO₂, indicating a highly dispersed structure. We are sorry that our XPS data brought much confusion to the reviewer. Moreover, we agree with the reviewer that XPS results of fresh samples cannot reflect the oxidation states of the active catalysts during reactions, but maybe different structures of Ir catalysts before n-butane dehydrogenation (BDH). Herein, we characterized the Ir₁/ND@G right after 0.5h in BDH to verify the catalytically active structure. As shown in **Supplementary Figure 18**, the Ir species are atomically dispersed on nanodiamond@graphene (ND@G) surface, indicating that the isolated Ir sites were the real active species during the reaction process. Combined with the stable theoretical model, we claim that single Ir atoms can represent the active catalysts. Following the suggestion, we added the figure and corresponding description in the revised manuscript (please see Line 224-226 on Page 14-15) and SI (please see **Supplementary Figure 18**).

Supplementary Figure 6. Ir L₃ XANES spectra of Ir₁/ND@G, Ir foil and IrO₂

Supplementary Figure 18. HAADF-STEM images of Ir₁/ND@G-used 0.5h

b. The EXAFS of the Ir₁ catalyst is similarly noisy. The EXAFS at $k >$ than about 7 \AA^{-1} is unreliable, which makes fitting for the FT inaccurate, see figure S8a and Table S2.

Response: Thanks for this precious suggestion. For the sample with low loading, the signal from single Ir atoms is weak and difficult to detect. We carefully reevaluated the quality of the XAS data of the Ir₁ catalyst. Here, we update a new graph of the XAS result in the revised SI to demonstrate that the quality of the XAS

data is enough for the fitting (**Supplementary Figure 9**). The EXAFS at k (from 1 \AA^{-1} to 10 \AA^{-1}) is reliable according to the new graph. And the parameters for the fitting results (i.e., σ^2 , E_0 shift and R-factor) as shown in **Supplementary table 2** fall within reasonable ranges, consistently proving that our fitting is accurate and reliable. We are also sorry that the previous version of the XAS results in SI caused some confusion.

Supplementary Figure 9. EXAFS fitting results of Ir L_3 -edge in k^3 space for (a) $\text{Ir}_1/\text{ND}@G$, (b) $\text{Ir}_{1+n}/\text{ND}@G$, (c) $\text{Ir}_n/\text{ND}@G$ and (d) $\text{IrNPs}/\text{ND}@G$

c. The STEM shows isolated Ir, at least when exposed to air. Are catalysts after butane dehydrogenation still isolated sites. The samples should be run in an eTEM cell, so that they can be treated in 450°C in H_2 and then analyzed without exposure to air?

Response: We appreciate the reviewer for this nice suggestion. To avoid the unnecessary exposure to air, all samples were sealed within He-filled containers as soon as synthesized to preserve their intrinsic structures before they were loaded into TEM's high-vacuum chamber for the characterization. We agree that in-situ TEM is a powerful tool to verify the real structure of active species under reaction atmosphere. However, the short-time exposure to air during the complex sample loading is still inevitable. Furthermore, due to the dramatically increased electron scattering from eTEM cell membranes (usually two 50nm-thick SiN films both above and beneath the 5-6nm-thick nanodiamond sample) and from the H_2 atmosphere, as well as the thermal turbulence in 450°C high temperature, we are afraid that the originally weak signal of single Ir atoms cannot be detected, considering our equipment capacities. Alternatively, we characterized the $\text{Ir}_1/\text{ND}@G$ right after 0.5h in BDH to verify the catalytically active structure. As shown in **Supplementary Figure 18**, the Ir species are all isolated Ir atoms without any Ir clusters or Ir NPs, indicating that the isolated Ir sites were the real active species during the reaction process. Following the suggestion, we added the figure and corresponding description in the revised manuscript (please see Line 224-226 on Page 14-15) and SI (please see **Supplementary Figure 18**).

d. Until the exact structure and oxidation state of the Ir1 is determined, the results from the DFT are not reliable. What if these are very small metallic Ir clusters under reaction conditions? The structure of the isolated pre-catalyst is not representative of the catalytic properties.

Response: We appreciate the reviewer for this nice comment. The uniform distribution of single Ir atoms in Supplementary Figure 18 after exposed in reaction atmosphere demonstrates that the isolated Ir sites represent the real catalytic structure in Ir₁/ND@G. The theoretical model of Ir-C₃ is then reasonable for investigating the Ir BDH nature of structure-dependence and metal-dependence.

Regarding the possible slight agglomeration of Ir atoms after long-time reaction, as shown in Figure 3(a), the structural evolution from single Ir atoms to clusters leads to depressed conversion rates. Through comparison of the TOF values between Ir₁/ND@G and Ir_{1+n}/ND@G, it is evident that the formation of small clusters is the main reason for deactivation of Ir₁/ND@G. Single Ir atom catalyst has higher activity than those of Ir clusters and nanoparticles, even if a slight agglomeration after the long-time reaction. The slight agglomeration actually did not hinder us to disclose the Ir BDH nature of structure-dependence and metal-dependence. Because we chose the reaction data point at 0.35h (the first point we measured) to calculate the butane conversion and TOF values, thereby suppressing the influence of possible structural evolution after long-time reaction.

The characterizations of the novel Ir1 catalysts fail to define the active oxidation state and coordination geometry.

Response: The fresh Ir₁/ND@G could not represent the real active catalyst during reaction, but as discussed above, the 0.5h-used sample shows similar structure to that of fresh one. Therefore, we believe that our current data can support the claim of isolated Ir site. We are sorry to bring much confusion to the reviewer. Following the suggestion, we added the experimental results and discussion of Ir₁/ND@G after 0.5h reaction in the revised manuscript (please see Line 224-226 on Page 14-15) and SI (please see **Supplementary Figure 18**).

2. The catalytic performance is incomplete. All Ir catalysts appear to have high C₄ olefin selectivity of greater than about 98%. The selectivity is typically lower at higher conversions; thus, these need to be measured at different conversions and compared at the same selectivity. Commercial catalysts operate at 40% propane conversion (at 1 atm C₃). What is the selectivity of each catalyst at these conversions? For dehydrogenation catalysts cycle length and regenerability are key metrics. What is the deactivation rate (and cycle length) for these catalysts? Can these be regenerated? Can the C and/or diamond support be regenerated?

Response: We thank the reviewer for this highly constructive suggestion. Herein, we increased the mass of catalysts to calculate both C₄ olefin selectivity and deactivation rate constant K_d at higher conversion, with the detailed catalytic performance shown in **Supplementary Figure 12** and **Supplementary Table 4**. All Ir catalysts showed high C₄ olefin selectivity (~95%). Especially for Ir₁/ND@G, the C₄ olefin selectivity was as high as 99% at the doubled conversion.

We agree with the reviewer that catalysts cycle length and regenerability are also key metrics in alkane dehydrogenation. The ND@G-supported metal catalysts have been successfully regenerated in our previous work¹. As shown in the TG-MS profile, a sharp decrease started to occur at the temperature of 430 °C (**Figure R1**), indicating the combustion of the nanodiamond in oxidation atmosphere². The combustion of coke was below 400 °C. Thus, we carried out the oxidation treatment at 300 °C, followed by a reduction process under the H₂ flow. As shown in **Supplementary Figure 20**, both butane conversion and butene selectivity were almost restored for two regeneration cycles, while the K_d values were 0.0228 h⁻¹ and 0.0216 h⁻¹ respectively.

The butene selectivity was above 94% throughout the reusability test. Therefore, we would like to claim that our Ir catalysts are regeneratable, with inevitable slightly declined activity after cycles, which may result from the formation of clusters.

We thank the reviewer for this valuable suggestion on the stability and reusability of the catalyst, which helps to substantially strengthen our manuscript. According to the suggestion, we have added above results and discussion about selectivity and reusability of Ir catalysts in the revised manuscript (please see Line 192-196 on Page 13 and Line 230-234 on Page 15) and SI (please see **Supplementary Figure 12** and **20** and **Supplementary Table 4**)

Supplementary Figure 12. n-Butane conversion and butene selectivity by time-on-stream during BDH at 450 °C, GHSV = 18000 mL · g_{cat}⁻¹ · h⁻¹, n-C₄H₁₀ : H₂ = 1:1 with He balance.

Supplementary Table 4. Catalytic performance of the catalysts for BDH

Catalysts	Conversion (%)	Selectivity (%)			K _d (h ⁻¹)
		C ₄ olefins	Butene	Others	
Ir ₁ /ND@G	13.6/10.2	99.0/99.3	94.8/94.9	4.1/4.3	0.0326
Ir _{1+n} /ND@G	26.8/19.0	97.3/98.7	93.7/95.3	4.0/3.4	0.0445
Ir _n /ND@G	30.4/15.9	94.7/98.5	89.5/94.2	5.3/4.2	0.0837
IrNPs/ND@G	19.3/10.5	96.3/98.0	91.0/93.0	5.3/5.0	0.0712

Figure R1. TG-MS profile of ND@G

Supplementary Figure 20. The conversion of butane and selectivity of butene over Ir₁/ND@G catalyst after several regeneration cycles. Reaction condition: atmospheric pressure, GHSV = 30000 mL · g_{cat}⁻¹ · h⁻¹, C₄H₁₀:H₂ = 1:1, He balance, 450 °C.

3. Single site catalysts on graphite and N-doped graphite are well known in electrochemistry for metals like Co, Ru, Fe and others. The metal loadings of these are often > 1% and display unique catalytic properties compared to metallic NPs. The Ir₁ catalyst has 0.02% Ir. Is this the maximum loading? Even if the TOF is higher than Ir NP, the rate/g will be low. If these can't be prepared at higher loading, it is unlikely that these will be used for any practical reaction.

Response: Thanks for this comment. Limited by the wet chemistry synthesis method, the maximum Ir loading is 0.1wt%, confirmed by ICP-OES. In STEM images of 0.1wt% Ir/ND@G, uniformly dispersed Ir single atoms are clearly observed without any Ir clusters or Ir NPs (**Figure R2**). The well-dispersed Ir atoms are highlighted by red circles. In Ir L₃-edge EXAFS spectra, no Ir-Ir bonding was observed (**Figure R3**), indicating all Ir atoms were atomically dispersed on ND@G support. In principle, the Ir₁/ND@G with a low Ir loading has a high TOF, but a low conversion. So, we have to agree with the reviewer that the current catalyst would not be suitable for practical applications. However, benefited from the homogeneous distribution and the simple model of Ir₁/ND@G, it is a suitable catalyst for fundamental research, such as exploring the catalytic nature of metal-dependence and structure-dependence in our manuscript. Using Ir₁/ND@G as a model catalyst, we found that the highly efficient catalyst is not monotonically determined by an increasingly unsaturated structure but a moderate strength of adsorption, which also strongly couples to the metal species. Due to the moderate strength of adsorption, isolated metal sites, instead of ensemble sites, favor both reactant adsorption and activation, leading to unparallel catalytic performance. Such fundamental insight on the BDH catalytic mechanisms and, more generally, the catalyst design guideline forms the critical discovery of our work, using Ir₁/ND@G as the model system.

To clarify the loading capability and strengthen our paper central topic, we added respective discussion in the revised manuscript (please see Line 383-384 on Page 24).

Figure R2. HAADF-STEM images of 0.1wt% Ir/ND@G

Figure R3. k^3 -weighted EXAFS spectra of 0.1wt% Ir/ND@G, Ir foil and IrO₂

Preparation of single site catalysts with different properties, which are different from metallic NPs is well known for several metals. These occur on defect sites on C and N-doped C, similar to those described here. The characterizations also need to be obtained under in situ conditions similar to butane dehydrogenation to confirm that the as synthesized structure is the same as the catalytic structure. The accuracy and precision of these in situ characterizations also needs significant improvement to define both the local coordination geometry (number of bonds and bond distance) and active oxidation state. Then the modeling needs to reflect these active site structures, which may be different from those currently modeled.

This paper is not recommended for publication and should be sent to a more specialized catalysis journal after additional characterizations are completed.

Response: We thank the reviewer for the valuable suggestion on the structure of active sites. We agree that in-situ TEM is a powerful tool to verify the real structure of active species under reaction atmosphere. However, due to the dramatically increased electron scattering from gas-cell membranes (usually two 50nm-thick SiN films both above and beneath the 5-6nm-thick nanodiamond sample) and from the H₂ atmosphere, as well as the thermal turbulence in 450°C high temperature, we are afraid that the originally weak signal of single Ir atoms cannot be detected, considering our TEM capacities. For in-situ XAFS, the low loading amount (0.02wt%) and the harsh reaction condition (450°C) also lead to a noisy and weak signal of single Ir atoms. It is difficult to detect and analyze the precise structure information. Therefore, current eTEM and in-situ XAFS techniques cannot provide the solid but desired evidence on the structure of Ir₁/ND@G under reaction conditions. To verify the catalytic structure, we carefully examined the HAADF-STEM images of the Ir₁/ND@G after 0.5h BDH at 450 °C and confirmed the absence of Ir clusters and small nanoparticles (**Supplementary Figure 18**). Therefore, we claim that single Ir atoms can represent Ir₁/ND@G catalyst and Ir-C₃ is a reasonable catalytic model to investigate the reaction mechanism.

To clarify the challenges for in-situ characterization techniques, we added some discussion in the revised **Method** section (please see Line 459-469 on page 27-28)

Reviewer #2 (Remarks to the Author):

The work by Ma et al. on alkane dehydrogenation is a novel study of metal and structure dependence of activity and selectivity for this reaction. Some points need to be clarified after publication in NC:

We thank the reviewer for the very positive view of our work. All the comments make the manuscript stronger. We have addressed all the comments point by point and revised the manuscript accordingly

1. The reaction rate almost decreases to half of the initial value after 10 h (Fig. 4) or 20h (Fig S11). Longer TOS values are needed to check for the deactivation of the active sites.

Response: We thank the reviewer for this valuable suggestion. As suggested, a long-term stability test for 50h has been carried out over Ir₁/ND@G (**Supplementary Figure 14**). The n-butane conversion rate was 6.8% and a high activity level of 4.1% was kept after 50 h reaction. The selectivity towards butene was as high as 95.7% after 50 h reaction, and the K_d value was only 0.0107 h⁻¹. Furthermore, we also investigated the reusability of Ir₁/ND@G catalysts through repeated regeneration processes (**Supplementary Figure 20**). The catalytic activity has been recovered, indicating that most of isolated Ir atoms were robust and active after regeneration. Following the suggestion, we have added the experimental results and corresponding description of the 50h stability test in the revised manuscript (please see Line 200-203 on Page 13) and SI (please see **Supplementary Figure 14**).

Supplementary Figure 14. Stability test over Ir₁/ND@G at 450 °C, GHSV = 30000 mL · g_{cat}⁻¹ · h⁻¹, n-C₄H₁₀:H₂ = 1:1 with He balance.

2. Experimental characterization of the sample after such TOS is needed to confirm the robust, atomically dispersed, and isolated nature of the active site.

Response: We appreciate the reviewer for the nice suggestion. As shown in **Supplementary Figure 16**, the majority of Ir species after 10h BDH reaction was still atomically dispersed as single atoms, showing few clusters of only 2 or 3 Ir atoms. This result demonstrates that the isolated Ir sites were robust even after 10-hour BDH. However, after 20-hour BDH, a small amount of Ir atoms aggregated into ensemble sites. Some isolated Ir atoms have been still found on ND@G (**Supplementary Figure 19**), highlighted by red circles.

We notice that tiny Ir clusters inevitably began to emerge and to be the active sites after 20-hour reaction in the reductive atmosphere. This structural evolution is one of reasons for the catalyst deactivation. To avoid the influence of such structural evolution, we chose the conversion at 0.35h (the first point we measured) to calculate butane conversion rates and TOF values. Moreover, the STEM results of Ir₁/ND@G after 0.5h BDH (**Supplementary Figure 18**) demonstrate the atomically dispersed and isolated Ir structure is the active site for the achieved high performance. Therefore, we conclude that the active site in Ir₁/ND@G was isolated Ir, accounting for the discovered metal-dependence and structure-dependence.

Following the suggestion, we added the STEM images and corresponding description of Ir₁/ND@G after 20-hours stability tests in the revised manuscript (please see Line 226-229 on Page 15) and SI (please see **Supplementary Figure 19**). We thank the reviewer for pointing out the further improvement direction for our future work. Thermostability is crucial to evaluate the catalyst for alkane dehydrogenation. The more stable single atom catalyst has been desired for BDH.

Supplementary Figure 19. HAADF-STEM images of Ir₁/ND@G after 20h BDH.

3. Comparison with TON and TOF values from the literature, as well as selectivities and stability is required.

Response: We thank the reviewer for this nice suggestion. Following the suggestion, we compared the reaction condition, TOF value, selectivity and stability (K_d) of different supported metal catalysts in literature with those of our Ir catalysts (**Supplementary Table 6**). Among these catalysts, the Ir₁/ND@G showed high selectivity and stability simultaneously, because the moderate adsorption capability well suppresses side reaction and coke formation. Due to the lowest reaction temperature (450°C), the Ir₁/ND@G did not show the best activity among these catalysts. However, under the same condition, Ir₁/ND@G exhibits the highest activity (TOF value). We have added the comparison table and cited the related works in the revised SI (please see **Supplementary Table 6**).

Supplementary Table 6. Summary of the catalytic performance of various supported metal catalysts for BDH

Catalyst	Mass (g)	Temp. (°C)	Feed	GHVS (mL g ⁻¹ h ⁻¹)	TOS (h)	Conv. (%)	Butene Sele. (%)	TOF (s ⁻¹)	K_d (h ⁻¹)	Ref
Pd-Cu/Al	0.25	550	C ₄ H ₁₀ : H ₂ : N ₂ = 1:3:6	24000	10h	34/17	86.5	0.84	0.0922	3
Pd-Pt/Al						47/39.7	85.5	1.18	0.0297	4
PtSn/Sp-Zn-C	0.2	530	H ₂ /n- C ₄ H ₁₀ = 1.25	5400	2h	28/23	98/98	0.75	0.1319	5
Pt/B/SiO ₂	0.05	550	1% n- C ₄ H ₁₀ /Ar	2700	25h	1.6/1.5	75/75	0.034	0.0042	6
PtSn/MgAl ₂ O ₄	0.2	530	H ₂ /n- C ₄ H ₁₀ = 1.25	5400	2h	31/29	93.8/95	0.71	0.0476	7
PtSnIn(0.5)/γ- Al ₂ O ₃	0.2	530	H ₂ /n- C ₄ H ₁₀ = 1.25	5400	2h	30/28	95	-	0.0486	8
Pt-Sn/SiO ₂	0.05	550	20% C ₄ H ₁₀ /N ₂	24000	13h	33.6/26	99% (total C ₄)	9	0.0281	9
PtSn/Al ₂ O ₃	0.03	575	H ₂ :N ₂ :n- C ₄ H ₁₀ =1:1 :1	72,000	10h	37.3/33.8	91.7/91.3	10.2	0.0152	10
Pt/Mg(In)(Al) O	0.005	530	H ₂ /C ₄ H ₁₀ = 2.5	124800	2h	13/7.5	97/97	5.53	0.3057	11
Pt/Sn/Zn/γ- Al ₂ O ₃	0.5	550	C ₄ H ₁₀ : N ₂ =1:1	600	6h	76.0/59.1	62.4/79.1	-	0.1307	12
Pt-TS-1	0.1	500	C ₄ H ₁₀ : H ₂ : Ar = 5: 1: 94	1500	10h	19/20	97 (total C ₄)	0.193	-	13
PtMn/SiO ₂ - Reduction	0.005- 0.055	500	20% C ₄ H ₁₀ /N ₂	24000	100h	16/13	99	-	0.002	14

PtMn/SiO ₂ -RWGS						50/18	99	-	0.015	
Pt _{1.75n} /ND@G	0.05	450	C ₄ H ₁₀ :H ₂ = 1:1, He balance	18000	10h	35.4/30.9	96.6/96.7	0.12	0.0223	¹⁵
Ir ₁ /ND@G	0.02	450	C ₄ H ₁₀ :H ₂ = 1:1, He balance	45000	10h	4.6/3.5	95.6/96.2	0.48	0.0241	The work
Ir _n /ND@G						14.7/9.1	91.7/94.3	0.025	0.0749	
IrNPs/ND@G						6.0/4.3	90.7/94.3	0.014	0.041	

4. What is the role of either nanodiamond, graphene, or noble metal site?

Response: We thank the reviewer for this good question. When annealed at 1100°C, dangling bonds on the nanodiamond surface evolve into new π -bonds and eventually form few-layer graphene coating on the nanodiamond surface, denoted as nanodiamond@graphene (ND@G). The sp²-bonded graphene shell covers sp³-bonded carbon core. In the graphene shell, rich structural defects are energetically favorable for anchoring metal atoms and stabilizing atomically dispersed metal species through metal-C bonding. In current work, the noble metal site serves as the active site to adsorb butane and intermediates. The strong interaction between Ir and C atoms on ND@G can prevent Ir single atoms from aggregating into Ir clusters.

5. Substrate scope with different alkanes is suggested to find trends between substrate structure and catalyst performance.

Response: We appreciate the reviewer for this valuable suggestion. We agree with the reviewer that a substrate scope is important to check the conclusions of structure-dependence and metal-dependence. Following the suggestion, the catalytic performance of PDH was measured over these catalysts under atmosphere pressure at 500 °C with 50 mg catalysts. A gas mixture of 5% C₃H₈ with He balance (flow rate = 15 mL min⁻¹) was used. The effluent mixture gas was analyzed by on-line gas chromatography (Agilent 7890 with an FID and a TCD detector). The similar trend has been found. For the structure-dependence, Ir₁ND@G showed the higher activity than Ir_n/ND@G and IrNPs/ND@G (**Supplementary Figure 27**). For the metal-dependence, Ir₁ND@G showed a high activity. Pt₁/ND@G and Pd₁/ND@G were almost inactive for PDH (**Supplementary Figure 28**). Following the suggestion, we have added these PDH results and corresponding description in the revised manuscript (please see Line 337-342 on Page 21) and SI (please see **Supplementary Figure 27** and **28**).

Supplementary Figure 27. TOF and C₃H₆ selectivity over Ir₁/ND@G, Ir_{1+n}/ND@G, Ir_n/ND@G and IrNPs/ND@G.

Supplementary Figure 28. C₃H₈ conversion over Ir₁/ND@G, Pt₁/ND@G, Pd₁/ND@G and ND@G.

6. The scope of graphene and nanodiamond with different oxidation degrees will also unravel the role of the support and its acid-base properties, as well as metal-support interactions.

Response: We thank the reviewer for this nice comment. The O atoms on ND@G surface can stabilize Ir atoms by forming Ir-O-C bonds (**Figure R4**). In our following work, we doped O atoms onto the ND@G surface, where the oxygen content increased from 3.7% to 5.7%. Benefited from the abundant O sites, the maximum Ir loading amount can be doubled (**Figure R5**).

Figure R4. XPS spectra of ND@G and ND@G-O₂

Figure R5. (a-c) HAADF-STEM images of 0.22wt% Ir/ND@G. (d) k^3 -weighted EXAFS spectra of 0.22wt% Ir/ND@G, Ir foil and IrO₂

Reviewer #3 (Remarks to the Author):

The authors present an interesting study of highly dispersed Ir, including atomically dispersed Ir. The as-prepared catalysts are carefully characterized, and catalytic testing shows promising butane dehydrogenation rates and relatively good catalyst stability.

Most of the text attributes the high activity of the Ir/ND@G catalyst to the atomic dispersion. However, Figure S10 shows that after reaction, there isn't any evidence that single atoms are still present on the catalyst. There is some mention that the catalyst is covered in coke, but this isn't elaborated and whether this could affect the STEM detection of individual Ir atoms.

Response: We thank the reviewer for this nice comment. We agree with the reviewer that coking can cover the isolated Ir atoms and affect the STEM detection. Herein, we carefully checked the STEM images and highlighted isolated Ir atoms and clusters by yellow and red circles, respectively. As shown in **Supplementary Figure 16**, Ir species was mainly atomically dispersed after 10-hour BDH. We are sorry that we did not highlight single Ir atoms and thus bring some confusion to the reviewer. Following the suggestion, we provided the additional STEM images in the revised SI (please see **Supplementary Figure 16**).

Supplementary Figure 16. HAADF-STEM images of Ir₁/ND@G after 10h BDH

Ultimately this casts doubt into whether the active catalyst actually functions as single atoms. It is clear that it starts out that way, but perhaps it is all converted to the small Ir clusters seen in Figure S10b. It is definitely clear that synthesizing the catalyst as isolated Ir atoms leads to smaller Ir particles over time during reaction, but since the Iridium loading was smaller during these reactions, it isn't fully clear if this is due to the atomic dispersion or the lower loading overall.

Response: We thank the reviewer for this valuable comment. After 10h reaction, abundant Ir atoms were observed and highlighted by yellow circles in **Supplementary Figure 16**. In reduction atmosphere for 10h, it is reasonable for single atoms to aggregate into clusters (few red circles in **Supplementary Figure 16**) via Ostwald ripening. To investigate the Ir BDH nature of structure-dependence and metal-dependence, we lowered the Ir loading down to 0.02% to increase the average distance between two Ir atoms and used the data point at 0.35h reaction to calculate TOF and conversion rate, avoiding the influence of any possible Ir aggregation. In addition, STEM images of different magnifications showed that Ir species were atomically dispersed after 0.5 BDH (**Supplementary Figure 18**). Therefore, our above results consistently support that the active catalyst (Ir₁/ND@G) functioned as single atoms.

The catalyst synthesis and catalytic results are both of high quality and worthy of publication, but the claim that the active catalyst is an isolated atom requires some careful consideration before it can be made.

Response: We appreciate the reviewer for the positive view of our work. Following the suggestion, we have added experimental results and discussion of Ir₁/ND@G after 0.5h reaction to support the claim of isolated Ir site in the revised manuscript (please see Line 224-226 on Page 14-15) and SI (**Supplementary Figure 18**). We can see all Ir species were atomically dispersed without any clusters or nanoparticles. Therefore, we

concluded that isolated Ir atoms can represent the active sites in our topic catalyst. We appreciate all comments from the reviewer, which helps us to improve our manuscript.

Reference

1. Zhang, J. *et al.* Nanodiamond-Core-Reinforced, Graphene-Shell-Immobilized Platinum Nanoparticles as a Highly Active Catalyst for the Low-Temperature Dehydrogenation of n-Butane. *ChemCatchem* **10**, 520-524 (2018).
2. Liu, J. *et al.* Origin of the Robust Catalytic Performance of Nanodiamond Graphene-Supported Pt Nanoparticles Used in the Propane Dehydrogenation Reaction. *ACS Catal.* **7**, 3349-3355 (2017).
3. Saxena, R. & De, M. Ni/Cu/Ag promoted Pd/Al₂O₃ catalysts prepared by electroless co-deposition for enhanced butane dehydrogenation. *Mater. Chem. Phys.* **261**, 124236 (2021).
4. Saxena, R. & De, M. Enhanced performance of supported Pd-Pt bimetallic catalysts prepared by modified electroless deposition for butane dehydrogenation. *Appl. Catal. A Gen.* **610**, 117933 (2021).
5. de Miguel, S., Ballarini, A. & Bocanegra, S. New PtSn structured catalysts with ZnAl₂O₄ thin film for n-butane dehydrogenation reaction. *Appl. Catal. A Gen.* **590**, 117315 (2020).
6. Byron, C. *et al.* Role of Boron in Enhancing the Catalytic Performance of Supported Platinum Catalysts for the Nonoxidative Dehydrogenation of n-Butane. *ACS Catal.* **10**, 1500-1510 (2019).
7. de Miguel, S. R., Vilella, I. M. J., Zgolicz, P. & Bocanegra, S. A. Bimetallic catalysts supported on novel spherical MgAl₂O₄-coated supports for dehydrogenation processes. *Appl. Catal. A Gen.* **567**, 36-44 (2018).
8. Bocanegra, S., de Miguel, S., Zgolicz, P. & Ballarini, A. n-butane dehydrogenation on PtSnIn and PtSnGa trimetallic catalysts supported on structured materials prepared by washcoating. *Inorg. Chem. Commun.* **134**, 109033 (2021).
9. Deng, L. *et al.* Elucidating strong metal-support interactions in Pt-Sn/SiO₂ catalyst and its consequences for dehydrogenation of lower alkanes. *J Catal.* **365**, 277-291 (2018).
10. Natarajan, P.; Khan, H. A.; Yoon, S.; Jung, K.-D., One-pot synthesis of Pt-Sn bimetallic mesoporous alumina catalysts with worm-like pore structure for n-butane dehydrogenation. *J Ind. Eng. Chem.* **63**, 380-390 (2018).
11. Wu, J., Peng, Z., Sun, P. & Bell, A. T. n-Butane dehydrogenation over Pt/Mg(In)(Al)O. *Appl. Catal. A Gen.* **470**, 208-214 (2014).
12. Seo, H. *et al.* Direct dehydrogenation of n-butane over Pt/Sn/M/gamma-Al₂O₃ catalysts: Effect of third metal (M) addition. *Catal. Commun.* **47**, 22-27 (2014).
13. Shao, M., Hu, C., Xu, X., Song, Y. & Zhu, Q. Pt/TS-1 catalysts: Effect of the platinum loading method on the dehydrogenation of n-butane. *Appl. Catal. A Gen.* **621**, 118194 (2021).
14. Liu, Y. *et al.* Promoting n-Butane Dehydrogenation over PtMn/SiO₂ through Structural Evolution Induced by a Reverse Water-Gas Shift Reaction. *ACS Catal.* **12**, 13506-13512 (2022).
15. Zhang, J. *et al.* Tin-Assisted Fully Exposed Platinum Clusters Stabilized on Defect-Rich Graphene for Dehydrogenation Reaction. *ACS Catal.* **9**, 5998-6005 (2019).

REVIEWER COMMENTS

Reviewer #1 (Remarks to the Author):

The authors have taken a more critical look at the data quality of XPS and XAS, provided more details on the stability and regenerability of the catalysts. However, there are still significant details that are not sufficiently discussed before this paper can be accepted. Critical analysis of the XAS data, for example, may establish that the single site Ir may not be single site Ir⁴⁺ as suggested. Rather under reaction conditions, this may be a very small metallic NP. In addition, because of the very low Ir loading of the single site catalyst, the elemental analysis may be inaccurate. If true, the catalytic rates may be similar to the other metallic catalysts. Thus, I do not recommend this paper until these issues are better quantified.

Comments:

1. In the authors' response letter, they admit that the catalyst stability decreases with time, regeneration above about 400C will oxidize the support, and the Ir loading is too low to make a practical catalyst. Instead, they suggest that the single site Ir catalyst will be of fundamental interest for other studies. However, in the Abstract the implication is that these catalysts solve many of the previous catalyst deficiencies and will be industrially applicable. In addition, the first paragraph of the introduction describes the commercial importance of alkane dehydrogenation for production of olefins. The paper, leads the reader to expect that this paper will describe a technically important catalyst. Furthermore, if these catalysts are only of fundamental importance, the a few examples of how these materials might address fundamental issues for dehydrogenation, or other reactions, should be discussed. The goals and importance of this work needs to be more clearly aligned with the potential the single site catalyst.

2. The introduction describes homogenous Ir catalysts and their limitations for alkane dehydrogenation. This is misleading. Homogenous catalysts are not capable of alkane dehydrogenation. These are stable only at temperatures below about 100C, where the alkane olefin equilibrium is too low for practical alkane dehydrogenation processes. The introduction is about hydrogen transfer from one alkane to an olefin. While this reaction is of some fundamental interest, this does not address important fundamentals of heterogeneous alkane dehydrogenation catalysts, which operate in the gas phase above about 500C. The introduction needs to address alkane dehydrogenation of heterogenous catalysts and how this paper relates to that literature. That introduction needs to define critical knowledge gaps and how this study addresses those.

3. At the end of the Introduction (lines 92-93) it is suggested that single site Ir will be compared to single site Pt and Pd catalysts. These comparisons are not made. There are no results, or discussion from the literature, for these. In addition, single site Pt and Pd are not stable against reduction to metallic NPs at alkane dehydrogenation conditions. How are these to be compared? Single site Pt is known in the literature. If the Ir are stable against reduction at dehydrogenation conditions, why are Pt and Pd different from Ir. For standard NP catalysts, each reduces at approximately the same temperature,

which is about 200C.

4. In line 106, it is suggested that the nano-diamonds (ND) have unique and abundant defect sites to stabilize metal atoms. This not correct. Graphite, N-doped carbon, high surface area oxide supports all have defect sites, which stabilize single site metal (ions). Furthermore, the ND does not have many sites. The maximum Ir loading is 0.02 wt%, which is much lower than most other supports. This low loading alone is a major barrier for practical applications, and even many fundamental studies.

5. The technical results also need significantly improvement to confirm the single site structure under reaction conditions. While the oxidized STEM is convincing of low nuclearity Ir species, the XAS under reaction is not. Small metallic group 8 metals oxidize upon RT exposure to air, even for short times. Clusters of a 10's atoms are nearly completely oxidized. The experimental details of the XAS data collection are not given in sufficient detail to confirm that these catalysts were reduced, or reacted with butane, at reaction temperature, AND not exposed to air prior to the EXAFS data acquisition. If under reaction, there are small Ir NPs and the sample is exposed to air, then the EXAFS would be IrO₂. The EXAFS for the other samples that are thought to be metallic Ir NP but have a significant Ir-O CN, leading me to believe that EXAFS of all samples have been exposed to air. It is critical to obtain the in-situ data. The samples can be reduced at 450C and data taken at RT to get acceptable data quality; however, this MUST be in an air tight cell with no exposure to air.

6. The XANES of IrO₂, Ir foil and the single site Ir is plotted in Fig. S8. The Ir foil is not correct. Ir foils are thick and not possible to get in transmission. Fluorescence data of the commercial Ir foil has serious self-absorption effects, which strongly distorts the spectra. Fully reduced Ir NPs have a much higher white line than shown in figure S6. The IrO₂ and single site Ir/ND is poorly plotted to allow for determination of differences in the energy and shape of the XANES. If there is any small amount of metallic Ir in the (air exposed?) sample, there will be a small shift to lower energy in the leading edge of the XANES. From figure S6, this appears to be possible, but the resolution of the figure doesn't allow for careful comparison. The first derivative of the XANES might show a small metallic feature. I have obtained XANES and EXAFS of many single site ions on C, and oxide supports. The XANES is always very different from the thermodynamically stable oxide, e.g., IrO₂ in this study. Figure S6 shows that the XANES of IrO₂ and the single site catalyst are almost identical, which suggests these are small IrO₂ clusters. If so, these result from oxidation of small metallic Ir NPs.

7. The EXAFS in figure 2f shows Ir-O bonds, but also higher shell peaks (between 3-4 Å) that look like higher shell peaks in IrO₂. Single site Ir⁺⁴ ions would have weak Ir-O-C peaks and at a different distance. Also, the imaginary parts of the FT (which are not shown) would also be very different for these higher shell peaks. The higher shell peaks need to be fit to determine if these are due to Ir-O-Ir, or Ir-O-C.

8. The EXAFS in Fig 2f also appears to have a very small peak at about 2.75 Å, a very similar distance for metallic Ir. While it will be difficult to fit this small peak, if the catalyst is a small metallic NP and exposed to air, this level of metallic Ir is what one would expect. If the XAS on this single site Ir sample has been obtained after exposed to air, it is very likely that under reaction conditions, the catalytic structure is a

small metallic NP. Thus, the FFT modeling, and other conclusions are not supported by the data.

9. The XANES spectra of the other 3 Ir catalysts is not given. The EXAFS indicates these are a mixture of both metallic (Ir-Ir) and oxidized Ir (Ir-O).

10. The response letter suggested that the XPS data has been removed. The XPS spectra of the non-single site Ir are still present in the SI, figure S7, but is not discussed. From the EXAFS these are mixtures of both metallic and oxidized Ir. The spectra of Ir(1-n) is shifted to higher binding energy than the Ir(n) and Ir(NPs), yet the EXAFS of all 3 have metallic and Ir-O bonds. Why are these different?

11. The important result for catalysis is that the single site catalyst has a higher TOR compared to Ir NPs. The elemental analysis of the single site Ir is 0.02%. The error for this low loading is high. To get an accurate analysis, one would need to use 20-50g! Was this done? If not, the error in the elemental analysis could be very high, perhaps 2-4 times, and the TOR is not far off from the other samples. Also, the dispersion of all samples is assumed to be 1.0, Table S S1, but the differences in size might lead to additional errors of 2-4 times.

I am not convinced that the single site Ir are present under alkane dehydrogenation conditions. If so, why is this group 8 metal different from others, e.g., Pt and Pd? Before this paper can be accepted, the single site structure must be obtained in-situ. If these are single site Ir⁺⁴ ions, then why are these stable when other group 8 metals are not. I'm also not convinced that the TOR are exceptionally higher than metallic NPs. It is likely that the errors in elemental analysis and true dispersions account for the differences. Other single site ions have much lower TOR than metallic NPs. Why is Ir different from other single site ions? The theory calculations need to address these issues. The paper also needs to define the purpose of the paper, the knowledge gaps and clearly present what is new in this study. Finally, if these are single sited Ir⁺⁴ ions are suitable for study of new fundamental issues, what are those? A few examples to show why these materials are important would help the reader understand the impact of the paper. Based on these concerns, I do not recommend this paper.

Reviewer #2 (Remarks to the Author):

The authors have addressed all the comments and the manuscript is recommended for publication

Reviewer #3 (Remarks to the Author):

The authors have made good effort in responding to the reviewers' comments. I was pleased to see the additions to the manuscript regarding additional catalytic data and additional imaging establishing that atomically dispersed metal particles are still present following reaction. Systems such as these always

bring up more questions, but I feel that the results reported within the scope of the work performed are worthy of publication.

Reviewer #1 (Remarks to the Author):

The authors have taken a more critical look at the data quality of XPS and XAS, provided more details on the stability and regenerability of the catalysts. However, there are still significant details that are not sufficiently discussed before this paper can be accepted. Critical analysis of the XAS data, for example, may establish that the single site Ir may not be single site IR+4 as suggested. Rather under reaction conditions, this may be a very small metallic NP. In addition, because of the very low Ir loading of the single site catalyst, the elemental analysis may be inaccurate. If true, the catalytic rates may be similar to the other metallic catalysts. Thus, I do not recommend this paper until these issues are better quantified.

Comments:

1. In the authors' response letter, they admit that the catalyst stability decreases with time, regeneration above about 400C will oxidize the support, and the Ir loading is too low to make a practical catalyst. Instead, they suggest that the single site Ir catalyst will be of fundamental interest for other studies. However, in the Abstract the implication is that these catalysts solve many of the previous catalyst deficiencies and will be industrially applicable. In addition, the first paragraph of the introduction describes the commercial importance of alkane dehydrogenation for production of olefins. The paper, leads the reader to expect that this paper will describe a technically important catalyst. Furthermore, if these catalysts are only of fundamental importance, the a few examples of how these materials might address fundamental issues for dehydrogenation, or other reactions, should be discussed. The goals and importance of this work needs to be more clearly aligned with the potential the single site catalyst.

Response: We appreciate the reviewer for the nice suggestion. Following the suggestion, we carefully read and removed these words/sentences about industrial applications in the *Abstract* (please see Page 3 Line 31-32) and *Introduction* (please see Page 4 Line 45-48). We are sorry that our writing about large-scale industrial applications might bring some confusion to the reviewer. In *Introduction*, we added the sentence about the advantage of single atom catalysts (SACs) in fundamental research and cited the related works (please see Page 5-6 Line 85-94).

2. The introduction describes homogenous Ir catalysts and their limitations for alkane dehydrogenation. This is misleading. Homogenous catalysts are not capable of alkane dehydrogenation. These are stable only at temperatures below about 100C, where the alkane olefin equilibrium is too low for practical alkane dehydrogenation processes. The introduction is about hydrogen transfer from one alkane to an olefin. While this reaction is of some fundamental interest, this does not address important fundamentals of heterogeneous alkane dehydrogenation catalysts, which operate in the gas phase above about 500C. The introduction needs to address alkane dehydrogenation of heterogenous catalysts and how this paper relates to that literature. That introduction needs to define critical knowledge gaps and how this study addresses those.

Response: We appreciate the reviewer for the suggestion. As suggested, we addressed the heterogenous catalysts of alkane dehydrogenation and pointed out the challenge of designing Ir SACs in the revised manuscript (please see Page 5-6 Line 78-94).

We agree with the reviewer that homogenous catalysts are not capable of light alkane

dehydrogenation in the gas phase. But it should not be ignored that the pincer Ir complexes have been widely applied for the homogenous alkane dehydrogenation. In the **Reference 4**, the authors wrote “Kaska and Jensen were the first to report the use of a pincer-iridium complex ($(t^{Bu}4PCP)IrH_4$ (16- H_2) as a highly stable and robust catalyst for COA/TBE dehydrogenation. The complex 16- H_2 demonstrated excellent stability and activity at temperatures as high as 200 °C. The COA dehydrogenation catalyzed by 16- H_2 was performed under periodic addition of TBE (0.36 M), and the reaction proceeded at a rate of 12 Toh^{-1} giving a maximum of 1000 TONs”, demonstrating the high activity of C-H cleavage on Ir sites. “The last 2 decades have seen several modifications in the design of pincer-iridium framework with a quest to obtain improved activities in alkane dehydrogenation” demonstrated that Ir catalysts have been widely studied for the homogenous alkane dehydrogenation. Following the suggestion, we cited the above review in the revised manuscript (please see Page 4 Line 50).

3. At the end of the Introduction (lines 92-93) it is suggested that single site Ir will be compared to single site Pt and Pd catalysts. These comparisons are not made. There are no results, or discussion from the literature, for these. In addition, single site Pt and Pd are not stable against reduction to metallic NPs at alkane dehydrogenation conditions. How are these to be compared? Single site Pt is known in the literature. If the Ir are stable against reduction at dehydrogenation conditions, why are Pt and Pd different from Ir. For standard NP catalysts, each reduces at approximately the same temperature, which is about 200C.

Response: We appreciate the reviewer for this comment. Actually, we have compared the activity of $Ir_1/ND@G$, $Pt_1/ND@G$ and $Pd_1/ND@G$ in detail and discussed the metal-dependence of alkane dehydrogenation at atomic level in the manuscript. Please see the section in Page 18-23: **Metal-dependence of noble metal catalysts in alkane dehydrogenation**. $Pt_1/ND@G$ and $Pd_1/ND@G$ have been reported in our previous works for alkane dehydrogenation, such as butane and propane dehydrogenation. The structure and morphology have been also well-described there. Following the comment, we have supplementally cited the related works of $Pt_1/ND@G$ and $Pd_1/ND@G$ in the revised manuscript (Please see Page 18-19 Line 285-287). We agree with the reviewer that SACs are always prone to be unstable at alkane dehydrogenation conditions. To avoid the influence of structural evolution, we chose the conversion at 0.35h (the first point we measured) to calculate butane conversion rates and TOF values.

As shown in Figure 4, $Ir_1/ND@G$ showed the higher activity than $Pt_1/ND@G$ and $Pd_1/ND@G$. The reason for the distinct activity can be explained by the difference in calculated d-band centers, which for $Ir_1@Gr$, $Pt_1@Gr$ and $Pd_1@Gr$ are -2.64 , -4.09 and -4.13 eV, respectively (**Figure 5**). This result demonstrates that the stronger adsorption will occur on $Ir_1@Gr$ rather than $Pt_1@Gr$ and $Pd_1@Gr$, illustrating that the adsorption of intermediates is dependent on the metal species. As to the $Ir_1/ND@G$, $Pt_1/ND@G$ and $Pd_1/ND@G$, C-H activation is the rate-determining step. The stronger adsorption of intermediates results in the more stable transition states and lower energy barrier in the rate-determining step. Therefore, the activity of $Ir_1/ND@G$ was different from $Pt_1/ND@G$ and $Pd_1/ND@G$.

4. In line 106, it is suggested that the nano-diamonds (ND) have unique and abundant defect sites to stabilize metal atoms. This not correct. Graphite, N-doped carbon, high surface area oxide

supports all have defect sites, which stabilize single site metal (ions). Furthermore, the ND does not have many sites. The maximum Ir loading is 0.02 wt%, which is much lower than most other supports. This low loading alone is a major barrier for practical applications, and even many fundamental studies.

Response: Thanks for this comment. Actually, we have to say that the support we employed is not nanodiamond (ND), but the nanodiamond@graphene (ND@G). When annealed at 1100°C, dangling bonds on the ND surface evolve into new π -bonds and eventually form few-layer graphene coating on the ND surface. We have denoted the derivative as nanodiamond@graphene (ND@G). The sp^2 -bonded graphene shell covers the sp^3 -bonded carbon core. In the graphene shell, rich structural defects are energetically favorable for anchoring metal atoms and stabilizing atomically dispersed metal species through metal-C bonding, which is strongly confirmed by our previous work (Nat. Commun. 2021, 12, 2664; J. Am. Chem. Soc. 2022, 144, 3535-3542; Nat. Catal. 2022, 5, 485-493; ACS Catal. 2019, 9, 5998-6005; Nat. Commun. 2022, 13, 6798). In the manuscript, we have described our support ND@G to distinguish from ND and cited the related works. At Line 117-121 on Page 8, we wrote “ND@G, composed of the sp^3 diamond core and highly defective sp^2 graphene outer shells, has been used as the support (**Supplementary Figure 1**). The unique surface with abundant defects can trap and stabilize metal atoms by the metal-C bond. Its morphology and fine structure have been well studied in our previous reports¹⁶⁻¹⁸”.

We thank the comment about the maximum Ir loading. But we have to say that the maximum Ir loading is 0.1% and proved the atomically dispersed structure by STEM images (**Figure R1**) and Ir $L_{3\text{-edge}}$ EXAFS spectra (**Figure R2**).

Moreover, we believe that the catalysts of low metal loading are important model systems, which enable us to study the fundamental mechanisms of catalytic reactions. Here are some famous examples: (1) in a work by Corma et al., 0.03 wt% Pt/TiO₂ served as the model SAC to establish the relationships between structure and catalytic performance in the hydrogenation of 3-nitrostyrene, CO oxidation and photocatalytic H₂ evolution (ACS Catal. 2019, 9, 10626-10639). They compared the activity of single atoms and clusters/nanoparticles to identify the active sites during reactions. (2) Ma et al. also prepared 0.02 wt% Pt₁/ α -MoC as the model SAC for water-gas shift reactions (Nature 2021, 589, 396-401). While the Pt₁/ α -MoC has the highest intrinsic activity, the crowding surface Pt species can help to eliminate excess surface oxygen species, which is the main reason why a high stability can be observed in the Pt₁-Pt_n/ α -MoC catalyst. (3) Su et al. designed the Pt-CuO/CoAlO SAC with an ultra-low Pt loading (0.02 wt%) for the selective catalytic reduction of NO_x by CO (ACS Catal. 2023, 13, 224-236). They found that the negatively charged Pt has a stronger NO adsorption ability than the positively charged Pt, and Cu serves as the CO adsorption site. (4) Li et al. synthesized a highly efficient photocatalyst by assembling single Pt atoms (0.02 wt%) on a defective TiO₂ support (Angew. Chem. Int. Ed. 2020, 59, 1295 – 1301). They found that the presence of single Pt atoms promoted the construction of the Pt-O-Ti³⁺ atomic interface and facilitated the transfer efficiency of photogenerated electron from Ti³⁺ defective sites to single Pt atoms, thereby favoring the separation of electron-hole pairs and suppressing the subsequent recombination. This work presents an effective approach to boost catalytic performance via the construction of atomic interface. (5) Sun et al. found that only trace amounts of Rh (0.02 wt%) can generate active H atoms and facilitate the transformation from MoO₃ to MoO_xC_y intermediates,

which is the key route to α - MoC_{1-x} (J. Am. Chem. Soc. 2022, 144, 22589–22598). In this work, the authors established the relationship between the reduction behavior and the structural evolution to supply a feasible strategy for the α - MoC_{1-x} synthesis.

Therefore, to investigate the nature of structure-dependence and metal-dependence, we lowered the Ir loading down to 0.02% to increase the average distance between two Ir atoms, further avoiding the influence of any possible Ir aggregation. We agree that this low loading, even 0.1wt%, is a major barrier for practical applications and the SACs with a high metal loading remained desired in alkane dehydrogenation. But such catalysts of low metal loading are of fundamental importance as discussed above. We thank the reviewer for pointing out the further improvement direction for our future work. Following the comment, we have cited the typical works about low loading in the revised manuscript (Please see Page 26 Line 402-404).

Figure R2. HAADF-STEM images of 0.1wt% Ir/ND@G

Figure R3. k^3 -weighted EXAFS spectra of 0.1wt% Ir/ND@G, Ir foil and IrO_2

5. The technical results also need significantly improvement to confirm the single site structure under reaction conditions. While the oxidized STEM is convincing of low nuclearity Ir species, the XAS under reaction is not. Small metallic group 8 metals oxidize upon RT exposure to air, even for short times. Clusters of a 10's atoms are nearly completely oxidized. The experimental details of the XAS data collection are not given in sufficient detail to confirm that these catalysts were reduced, or reacted with butane, at reaction temperature, AND not exposed to air prior to the EXAFS data acquisition. If under reaction, there are small Ir NPs and the sample is exposed to air, then the EXAFS would be IrO₂. The EXAFS for the other samples that are thought to be metallic Ir NP but have a significant Ir-O CN, leading me to believe that EXAFS of all samples have been exposed to air. It is critical to obtain the in-situ data. The samples can be reduced at 450C and data taken at RT to get acceptable data quality; however, this MUST be in an air tight cell with no exposure to air.

Response: We appreciate the reviewer for this comment. We are so sorry that we didn't give the experimental details of the XAS experiments. We have updated the details in the revised manuscript (Please see Page 29 Line 472-474). For the quasi-*in situ* XAS spectra of reduced catalysts, the powdered sample was pressed into the sheet in the glovebox without exposure to air. The samples were measured in the fluorescence mode, using a Lytle detector to collect the data. The XAS samples were sealed in Kapton films with Ar protection after activation, and the whole process was performed in the glovebox. The reduced sample could not be oxidized by this sample preparation method for XAS data collection. The effectiveness of this method has been demonstrated in our previous studies (Nat. Commun. 2022, 13, 6798; J. Am. Chem. Soc. 2022, 144, 5108-5115; CCS Chemistry, 2022, 1-8).

Significantly, we have to emphasize that the existence of Ir-C/O bonds does not only imply the oxidized Ir species. By decreasing the size of supported metal particles, the fraction of metal-support interface substantially increased. The strong charge transfer between the metal particles and the support renders metal particles with higher electron-deficient charge states, which results in the formation of positively charged metal species (ACS Catal. 2020, 10, 11011-11045; ACS Cent. Sci. 2021,7,262-273).

It is critical to understand the structure of the catalyst during the reaction utilizing the *in situ* XAS method. However, *in situ* XAS experiments are challenging in our system because of the low metal loading (0.02 wt%). Herein, we provide STEM images of different magnifications (low and high magnifications) (**Supplementary Figure 18**), which strongly prove that Ir species are atomically dispersed after 0.5h BDH. These results demonstrate that the isolated Ir sites are the real active species during the reaction process. We acknowledged that it is reasonable for single atoms to aggregate into clusters via Ostwald ripening. As shown in **Figure 3a**, the formation of small clusters of 3-4 atoms is the main reason for the deactivation of Ir₁/ND@G. Single Ir atom catalyst has higher activity than those of Ir clusters and nanoparticles, even if there is slight agglomeration after the long-time reaction. To avoid the influence of such structural evolution, we chose the conversion at 0.35h (the first point we measured) to calculate butane conversion rates and TOF values, guaranteeing the real performance of Ir catalysts. Therefore, we conclude that the active site in Ir₁/ND@G is isolated Ir atom, accounting for the discovered metal-dependence and structure-dependence.

We are sorry that our results of Ir-C/O bonds in EXAFS brought some confusion to the reviewer. Following the suggestion, we added the discussion and cited typical papers in the revised manuscript (Please see Page 11 Line 162-165).

6. The XANES of IrO₂, Ir foil and the single site Ir is plotted in Fig. S8. The Ir foil is not correct. Ir foils are thick and not possible to get in transmission. Fluorescence data of the commercial Ir foil has serious self-absorption effects, which strongly distorts the spectra. Fully reduced Ir NPs have a much higher white line than shown in figure S6. The IrO₂ and single site Ir/ND is poorly plotted to allow for determination of differences in the energy and shape of the XANES. If there is any small amount of metallic Ir in the (air exposed?) sample, there will be a small shift to lower energy in the leading edge of the XANES. From figure S6, this appears to be possible, but the resolution of the figure doesn't allow for careful comparison. The first derivative of the XANES might show a small metallic feature. I have obtained XANES and EXAFS of many single site ions on C, and oxide supports. The XANES is always very different from the thermodynamically stable oxide, e.g., IrO₂ in this study. Figure S6 shows that the XANES of IrO₂ and the single site catalyst are almost identical, which suggests these are small IrO₂ clusters. If so, these result from oxidation of small metallic Ir NPs.

Response: We appreciate the reviewer for this comment. We got the Ir foil XAS spectra by transmission mode. As we know, all standard XAS spectra of all beamlines are collected by transmission mode. Highly dispersed Ir species displaying a similar oxide state with IrO₂ is probable (Angew. Chem. Int. Ed. 2022, 61, e202202654; Nat. Chem. 2021, 13, 887-894). Therefore, it is reasonable that the Ir₁/ND@G catalyst shows a similar oxide state with IrO₂ in our manuscript. We only observed the Ir-C/O coordination in the first shell in Fourier-transformed k^3 -weighted EXAFS spectra and no obvious Ir-O coordination in other coordination shells (**Figure 2d**), which demonstrates there are no IrO₂ clusters.

7. The EXAFS in figure 2f shows Ir-O bonds, but also higher shell peaks (between 3-4 Å) that look like higher shell peaks in IrO₂. Single site Ir⁺⁴ ions would have weak Ir-O-C peaks and at a different distance. Also, the imaginary parts of the FT (which are not shown) would also be very different for these higher shell peaks. The higher shell peaks need to be fit to determine if these are due to Ir-O-Ir, or Ir-O-C.

Response: Thanks for this comment. We can see that the actual intensity of higher shell peaks (between 3-4 Å) of the Ir₁/ND@G catalyst is small (**Figure 2d**). We only observed the Ir-C/O coordination in the first shell in Fourier-transformed k^3 -weighted EXAFS spectra and no obvious Ir-O coordination in other coordination shells (**Figure 2d**), which demonstrates there are no IrO₂ clusters. Therefore, we don't think there is Ir-O-Ir or Ir-C/O-C/O.

8. The EXAFS in Fig 2f also appears to have a very small peak at about 2.75 Å, a very similar distance for metallic Ir. While it will be difficult to fit this small peak, if the catalyst is a small metallic NP and exposed to air, this level of metallic Ir is what one would expect. If the XAS on this single site Ir sample has been obtained after exposed to air, it is very likely that under reaction conditions, the catalytic structure is a small metallic NP. Thus, the FFT modeling, and other conclusions are not supported by the data.

Response: We appreciate the reviewer for this comment. For the quasi-*in situ* XAS spectra of reduced catalysts in Figure 2d and Figure 2f, the powdered sample was pressed into the sheet in the glovebox without exposure to air. We can see that the actual total intensity of the Ir₁/ND@G catalyst is small, so the very small peak at about 2.75 Å, which is not the main coordination. (**Figure 2d**).

9. The XANES spectra of the other 3 Ir catalysts is not given. The EXAFS indicates these are a mixture of both metallic (Ir-Ir) and oxidized Ir (Ir-O).

Response: We have to emphasize again that the positively charged Ir^{δ+} species is not equal to IrO_x. The strong charge transfer between the metal particles and the support renders metal particles with higher electron-deficient charge states, which results in the formation of Ir-C/O bonds. From Ir₁/ND@G to Ir NPs/ND@G, the growing average size of the Ir species lead to the increased fraction of Ir-Ir bond.

10. The response letter suggested that the XPS data has been removed. The XPS spectra of the non-single site Ir are still present in the SI, figure S7, but is not discussed. From the EXAFS these are mixtures of both metallic and oxidized Ir. The spectra of Ir(1-n) is shifted to higher binding energy than the Ir(n) and Ir(NPs), yet the EXAFS of all 3 have metallic and Ir-O bonds. Why are these different?

Response: We thank the reviewer for this comment. The XPS spectra of Ir catalysts have been discussed in our manuscript. Please see Page 11 Line 167-170. We wrote “when the Ir loading increases, the Ir 4f_{7/2} peak shifted towards lower binding energy, indicating that Ir species were gradually close to the metallic state along with the formation of Ir clusters or nanoparticles (**Supplementary Figure 7**)”.

Significantly, we have to clarify that the positively charged Ir^{δ+} species is not equal to IrO_x. The highly dispersed Ir species bonded with C atoms on ND@G, thereby leading to the partial oxidation of Ir, similar to other metal-based catalysts (Nat. Catal. 2022, 5, 485-493; J. Am. Chem. Soc. 2022, 144, 5108-5115). Moreover, the CN of Ir_{1+n}/ND@G, Ir_n/ND@G and IrNPs/ND@G are 2.6, 3.5 and 3.6, respectively. The increased CN indicates the growing size and crystallization, which agrees well with our HAADF-STEM and CO-DRIFTS results. For Ir_{1+n}/ND@G, some Ir cluster (major species in the 0.53 nm range) has been observed together with abundant isolated Ir atoms, as shown in **Figure 1f, g** and **Supplementary Figure 2**. For Ir_n/ND@G, as the Ir loading increased, the island-like Ir clusters (major species in the 0.77 nm range) with 10-13 atoms have been well dispersed on ND@G, as shown in **Figure 1h, i** and **Supplementary Figure 3**. For IrNPs/ND@G, Ir nanoparticles (major species in the 1.72 nm range) are located on the ND@G surface predominantly, as shown in **Supplementary Figure 4**. Therefore, Ir_{1+n}/ND@G, Ir_n/ND@G and IrNPs/ND@G serve as the model for the mixture of single Ir atoms and clusters, the Ir clusters of 10-13 atoms and the Ir nanoparticles, respectively.

11. The important result for catalysis is that the single site catalyst has a higher TOR compared to Ir NPs. The elemental analysis of the single site Ir is 0.02%. The error for this low loading is high. To get an accurate analysis, one would need to use 20-50g! Was this done? If not, the error in the elemental analysis could be very high, perhaps 2-4 times, and the TOR is not far off from the other samples. Also, the dispersion of all samples is assumed to be 1.0, Table S S1, but the differences in

size might lead to additional errors of 2-4 times.

Response: We thank the reviewer for this comment about the error caused by low loading. We have to say, however, that a low loading amount does not mean a high error. In ICP-OES, Ir loading of 0.021% is much higher than the minimum detectable concentration (0.00015%). The 52 mg sample we used in test is enough for the accurate analysis by ICP-OES. Additionally, in the synthetic procedure, we have pre-prepared $\text{H}_2\text{IrCl}_6 \cdot 6\text{H}_2\text{O}$ solution (10g/L) and precisely extracted a certain amount of solution to add into ethanol. After adequately mixing, 200mg ND@G powder was added into the above solution. Here, we have raised the catalyst mass 10-fold to reduce the error caused by low loading. As mentioned, we have reduced the error to the minimum. The actual loading (0.021wt%) was much close to the theoretical value (0.02wt%).

On the other hand, the low metal loading is acceptable for investigating the fundamental catalytic mechanisms, as demonstrated by so many high-quality works. Here are some examples: (1) in a work by Corma et al., 10 mg 0.03 wt% Pt/TiO₂ has been used as the model SAC to establish the relationships between structure and catalytic performance of 3-nitrostyrene hydrogenation (ACS Catal. 2019, 9, 10626-10639). They compared the activity of single atoms and clusters/nanoparticles to identify the working sites during reactions. (2) Ma et al. prepared 0.02 wt% Pt₁/α-MoC as the model SAC for water–gas shift reactions (Nature 2021, 589, 396-401). While the Pt₁/α-MoC has the highest intrinsic activity, the crowding surface Pt species can help to eliminate excess surface oxygen species, which is the main reason why a high stability can be observed in the Pt₁–Pt_n/α-MoC catalyst. (3) Su et al. designed Pt–CuO/CoAlO SAC with an ultra-low Pt loading (0.02 wt%) for the selective catalytic reduction of NO_x by CO (ACS Catal. 2023, 13, 224–236). They found that the negatively charged Pt has a stronger NO adsorption ability than the positively charged Pt, and Cu serves as the CO adsorption site. (4) Li et al. synthesized a highly efficient photocatalyst by assembling single Pt atoms (0.02 wt%) on a defective TiO₂ support (Angew. Chem. Int. Ed. 2020, 59, 1295 – 1301). They found that the presence of single Pt atoms promoted the construction of the Pt-O-Ti³⁺ atomic interface and facilitated the transfer efficiency of photogenerated electron from Ti³⁺ defective sites to single Pt atoms, thereby favoring the separation of electron–hole pairs and suppressing the subsequent recombination. This work presents an effective approach to boost catalytic performance via the construction of atomic interface. (5) Sun et al. found that only trace amounts of Rh (0.02 wt%) can generate active H atoms and facilitate the transformation from MoO₃ to MoO_xC_y intermediates, which is the key route to α-MoC_{1-x} (J. Am. Chem. Soc. 2022, 144, 22589–22598). In this work, the authors established the relationship between the reduction behavior and the structural evolution to supply a feasible strategy for the α-MoC_{1-x} synthesis.

Therefore, the accurate analysis and activity tests on the sample with low loading (i.e., 0.02wt%) do not need 20-50 g for fundamental research. More importantly, our low metal loading is still high enough for modern analytic equipments, such as ICP-OES and STEM, so there is not a high error.

As for the comment about Ir dispersion, as shown in **Supplementary Table 1**, the Ir dispersion was not assumed but calculated by the equation: Ir dispersion=1/average particle diameter. Obviously, not all samples have the dispersion of 100%. The average particle diameter of clusters in Ir_{1+n}/ND@G or Ir_n/ND@G was less than 1 nm. Calculated by the equation, the Ir dispersion was 100%, indicating Ir species was atomically dispersed and fully exposed on ND@G. For

IrNPs/ND@G, the average particle diameter of Ir NPs was 1.72 nm. Calculated by the equation, the Ir dispersion was 58.1%. The difference in Ir dispersion has been considered in TOF calculation according to the equation (5) in page 27 of the manuscript. Therefore, the difference in size did not lead to additional errors in our manuscript.

Following the suggestion, the high-quality works about low loading have been cited in the *Method* (Please see Page 26 Line 402-404) and the discussion of Ir dispersion has been added in the revised SI (Please see Page 30).

I am not convinced that the single site Ir are present under alkane dehydrogenation conditions. If so, why is this group 8 metal different from others, e.g., Pt and Pd? Before this paper can be accepted, the single site structure must be obtained in-situ. If these are single site Ir+4 ions, then why are these stable when other group 8 metals are not. I'm also not convinced that the TOR are exceptionally higher than metallic NPs. It is likely that the errors in elemental analysis and true dispersions account for the differences. Other single site ions have much lower TOR than metallic NPs. Why is Ir different from other single site ions? The theory calculations need to address these issues. The paper also needs to define the purpose of the paper, the knowledge gaps and clearly present what is new in this study. Finally, if these are single sited Ir+4 ions are suitable for study of new fundamental issues, what are those? A few examples to show why these materials are important would help the reader understand the impact of the paper. Based on these concerns, I do not recommend this paper.

Response: We appreciate the reviewer for the comments. Herein, above main concerns have been responded as below.

1. **There is no IrO₂ on Ir₁/ND@G.** We have to say that the high oxidation state of Ir₁/ND@G resulted from the atomical dispersion of Ir species anchoring by Ir-C bonds, not from the formation of IrO₂ cluster or nanoparticle, as confirmed by EXAFS results. We have clearly explained in response to comment 6 and 7. The similar results have been also reported in many previous published works (Angew. Chem. Int. Ed. 2022, 61, e202202654; Nat. Chem. 2021, 13, 887-894).
2. **The nature of isolated Ir site.** For the quasi-*in situ* XAS spectra of reduced catalysts, the whole process was performed in the glovebox. Thus, the reduced sample could not be oxidized by the sample preparation method.
3. **There are no high errors in elemental analysis.** Ir loading of 0.021% is much higher than the minimum detectable concentration in ICP-OES. In the synthetic procedure and activity tests, we have reduced the error to the minimum. The actual loading (0.021wt%) was much close to the theoretical value (0.02wt%). Moreover, we also have to emphasize that low metal loading is widely adopted for fundamental studies. Many previous works reported important results using low metal loading (e.g. Angew. Chem. Int. Ed. 59, 2020, 1295-1301; Nature, 2021, 589, 396-401; ACS Catal. 2022, 13, 224-236; ACS Catal. 2019, 9, 10626-10639; J. Am. Chem. Soc. 2022, 144, 22589-22598).
4. **The reason for the high catalytic performance of Ir₁/ND@G.** Actually, in our manuscript, the higher catalytic performance of Ir₁/ND@G than Pt₁/ND@G and Pd₁/ND@G originated from their distinct d-band centers. The calculated d-band centers for Ir₁@Gr, Pt₁@Gr and Pd₁@Gr are -2.64, -4.09 and -4.13 eV, respectively, indicating that a stronger adsorption will occur on Ir₁@Gr than Pt₁@Gr and Pd₁@Gr. On catalysts of Ir₁/ND@G, Pt₁/ND@G and

Pd₁/ND@G, the C-H activation is the rate-determining step. The stronger adsorption of intermediates results in the more stable transition states and the lower energy barrier in the rate-determining step. Therefore, the activity of Ir₁/ND@G was different from that of Pt₁/ND@G and Pd₁/ND@G, demonstrating the metal-dependence in BDH. We have theoretically discussed why the Ir₁/ND@G is different from Pt₁/ND@G and Pd₁/ND@G in the section: **Metal-dependence of noble metal catalysts in alkane dehydrogenation**. The detailed discussion has been provided from reaction energies and d-band centers (please see Page 20-23 Line 313-360).

In our previous work, Pt₃ clusters with high ratios of unsaturated-coordinated atoms have a higher activity than single Pt atoms and Pt NPs. We also calculated the d-band center of Pt₃@Gr and found that the d-band center of Pt₃@Gr is -2.14 eV close to Ir₁@Gr (-2.64 eV). These results imply that the d-band centers of Pt₃@Gr and Ir₁@Gr are located at the region of high activity by realizing the balance of C-H activation and butene desorption. Moderate adsorption behavior is the key to the high catalytic activity and butene selectivity, which strongly depend on both metal structure and concentration.

Reviewer #2 (Remarks to the Author):

The authors have addressed all the comments and the manuscript is recommended for publication
Response: We appreciate the reviewer for the positive view of our work. All the comments help us to improve our manuscript.

Reviewer #3 (Remarks to the Author):

The authors have made good effort in responding to the reviewers' comments. I was pleased to see the additions to the manuscript regarding additional catalytic data and additional imaging establishing that atomically dispersed metal particles are still present following reaction. Systems such as these always bring up more questions, but I feel that the results reported within the scope of the work performed are worthy of publication.

Response: We appreciate the reviewer for the positive view of our work. All the comments make the manuscript stronger. Moreover, we specially thank the reviewer for pointing out the further improvement direction for our future work.

REVIEWER COMMENTS

Reviewer #1 (Remarks to the Author):

My major concern with this paper is that single site Ir catalysts for propane dehydrogenation (PDH) are suggested to be stable at reaction temperature, which is typically greater than 550C. My previous response was that unless the catalysts are treated under reaction conditions (propane or H₂) and the characterizations of the active site done in the absence of exposure to air, the analyses are compromised. My experience is that every group 8 metal is not stable on any support at 550C when there is H₂. Propane dehydrogenation even at 1% conversion gives enough H₂ to reduce ionic metals to metallic NPs. I have done, Pt, Pd, Rh, Ru and Ir on SiO₂, Al₂O₃, TiO₂, C, etc. None remains ionic under PDH conditions. In addition, all noble metallic catalysts surface oxidize upon exposure to air at RT. If the NPs are very small, ca. 1 nm, then every metallic atom becomes oxidized.

In the lengthy response to my questions, the authors cited their own previous papers as rebuttal to my comments. I don't know what was done in their previous studies, but I find the experimental details in this paper missing, thus I don't find the conclusions convincing. The recent changes to the manuscript are minimal. Thus, I am still not convinced the active site is a single ion site at catalytic conditions.

One of my questions was to give more details about the experimental conditions, so that I could judge whether these are the likely active site. The changes of the XAS experimental details that were added are: The samples were measured in fluorescence mode, using a Lytle detector to collect the data. The XAS samples were sealed in Kapton films with Ar protection after activation, and the whole process was performed in the glovebox. This does not clarify what was done, or if the data are reliable.

Were the catalysts reduced in the glove box, i.e., was the reactor and gases in the glove box? What was the propane concentration and conversion? Where the catalysts pretreated in H₂? Where the catalysts unloaded in the GB and pressed into XAS wafers and protected by Kapton? Where the samples stored on air tight sample vials in the GB prior to XAS measurement?

Alternatively, were the catalyst pretreated, or reacted, in the lab with a reactor with shut off valves at the inlet and outlet, which was transferred to the GB for preparation and storage of XAS samples?

Or was the catalyst reacted in the lab, cooled and unloaded in air, and quickly transferred to the GB? Such samples are always oxidized, which is what the current XAS and XPS appears to be consistent with. If this is the treatment, then the catalyst could be metallic under reaction conditions and oxidized when analyzed. If the metallic NP are very small, 100% of the metallic Ir would be oxidized, i.e., all surface atoms.

Without data that absolutely confirms that the catalytic Ir is ionic under reaction conditions, I will not be convinced. Furthermore, since all other group 8 metals do reduce, why is this catalyst different, i.e., why is it more stable against reduction. Single site Ir catalysts on C can be made and are reducible. What is

different here?

Without these new data and explanation of what is different, I do not need to re-review this paper. I do not recommend this manuscript. The other two reviewers find this work acceptable, thus, I'll leave it to the editor to determine the final decision.

Reviewer #1 (Remarks to the Author):

My major concern with this paper is that single site Ir catalysts for propane dehydrogenation (PDH) are suggested to be stable at reaction temperature, which is typically greater than 550C. My previous response was that unless the catalysts are treated under reaction conditions (propane or H₂) and the characterizations of the active site done in the absence of exposure to air, the analyses are compromised. My experience is that every group 8 metal is not stable on any support at 550C when there is H₂. Propane dehydrogenation even at 1% conversion gives enough H₂ to reduce ionic metals to metallic NPs. I have done, Pt, Pd, Rh, Ru and Ir on SiO₂, Al₂O₃, TiO₂, C, etc. None remains ionic under PDH conditions. In addition, all noble metallic catalysts surface oxidize upon exposure to air at RT. If the NPs are very small, ca. 1 nm, then every metallic atom becomes oxidized.

In the lengthy response to my questions, the authors cited their own previous papers as rebuttal to my comments. I don't know what was done in their previous studies, but I find the experimental details in this paper missing, thus I don't find the conclusions convincing. The recent changes to the manuscript are minimal. Thus, I am still not convinced the active site is a single ion site at catalytic conditions.

One of my questions was to give more details about the experimental conditions, so that I could judge whether these are the likely active site. The changes of the XAS experimental details that were added are: The samples were measured in fluorescence mode, using a Lytle detector to collect the data. The XAS samples were sealed in Kapton films with Ar protection after activation, and the whole process was performed in the glovebox. This does not clarify what was done, or if the data are reliable.

Were the catalysts reduced in the glove box, i.e., was the reactor and gases in the glove box? What was the propane concentration and conversion? Where the catalysts pretreated in H₂? Where the catalysts unloaded in the GB and pressed into XAS wafers and protected by Kapton? Where the samples stored on air tight sample vials in the GB prior to XAS measurement?

Alternatively, were the catalyst pretreated, or reacted, in the lab with a reactor with shut off valves at the inlet and outlet, which was transferred to the GB for preparation and storage of XAS samples?

Or was the catalyst reacted in the lab, cooled and unloaded in air, and quickly transferred to the GB? Such samples are always oxidized, which is what the current XAS and XPS appears to be consistent with. If this is the treatment, then the catalyst could be metallic under reaction conditions and oxidized when analyzed. If the metallic NP are very small, 100% of the metallic Ir would be oxidized, i.e., all surface atoms.

Without data that absolutely confirms that the catalytic Ir is ionic under reaction conditions, I will not be convinced. Furthermore, since all other group 8 metals do reduce, why is this catalyst different, i.e., why is it more stable against reduction. Single site Ir catalysts on C can be made and are reducible. What is different here?

Without these new data and explanation of what is different, I do not need to re-review this paper. I do not recommend this manuscript. The other two reviewers find this work acceptable, thus, I'll leave it to the editor to determine the final decision.

Response: Many thanks to the Reviewer #1 for continuous efforts, which helps us to further improve the manuscript. As for the Reviewer's concerns about the stability of the Ir₁/ND@G catalyst, this catalyst would be a mixture of single atoms and clusters (or small particles) in the long term reaction according to the STEM results (Supplementary Figure 16 and 18). The supplementary *in-situ* CO-DRIFTS experiments demonstrate the atomic dispersed Ir catalyst could reserve its structure at the initial period of the reaction. Therefore, the initial activity was utilized to compare the catalytic performance of single atoms, clusters, and nanoparticles, as well as different metal single atom catalysts. The single-atom Ir catalyst has the best catalytic performance for the butane dehydrogenation reaction, and developing construction methods of the stable single-atom Ir catalyst are pivotal for BDH in the future. Based on the above, we carefully revised the part of "Structure-dependence of Ir catalyst in alkane dehydrogenation" in the manuscript according to the suggestion of the Reivewer #1.

Before a point-to-point response, we kindly clarify that all the Ir catalysts in this work are designed for butane (C₄H₁₀) dehydrogenation not propane dehydrogenation the Reviewer #1 mentioned and the reaction temperature for the butane dehydrogenation is 450 °C not 550 °C the Reviewer #1 mentioned. All above main concerns from the Reviewer #1 have been responded to as below.

1. In above comment, the reviewer wrote his/her experience that every group 8 metal is not stable on any support at 550 °C when there is H₂. Actually, many stable single atom catalysts have been reported. (1) In recent work (Nat. Catal. 2022, 5, 1145–1156), Tao Zhang et al. developed Ru single atom catalyst with isolated Ru atoms supported on nitrogen-doped carbon for propane dehydrogenation (PDH). They found that the Ru species remained almost unchanged compared with the fresh catalyst after reduction treatment by H₂ at 600 °C or during the PDH reactions, indicating that Ru species remained atomically dispersed during PDH. (2) Similarly, Jeffrey T. Miller et al. developed an γ -Al₂O₃ supported isolated Ni (II) site by anchoring Ni²⁺ cations into Al³⁺ vacancy on γ -Al₂O₃ as a catalyst for PDH (ACS Catal. 2022, 12, 12607–12616). They found that the nature of the Ni sites remains constant for the fresh sample, regardless of 1h treatment by 3% H₂ at 600 °C or 3% C₃H₈ at 580 °C or 20% O₂ at 600 °C. This result suggests that the atomically dispersed Ni²⁺ sites retain the local structure and no reduction or aggregation occurs over reduction or reaction-regeneration cycles. (3) De Chen et al. developed Pt catalysts in a full-size range (from a single atom and sub-nanometer clusters to nanoparticles) and investigated the nature of size-dependence in PDH (reaction condition: T = 575 °C, H₂/C₃H₈=0.8 v/v, Ar balance) (ACS Catal. 2020, 10, 12932–12942).

Above typical works demonstrate that not all the single atom catalysts with group 8 metal on any support are unstable at 550°C when there is H₂. We need to emphasize that we used the initial reaction rate to compare the performance of different single atom catalysts in this manuscript.

2. In the comment, the reviewer claimed that the surface of all noble metallic catalysts is oxidized upon exposure to air at RT. In our previous work, Pt catalyst with small Pt NPs anchored on ND@G by Pt-C bonds has been developed (ACS Catal. 2017, 7, 3349–3355). In *ex-situ* XPS spectra, the Pt 4f7/2 peak was located at 70.8 eV, corresponding to metallic Pt (70.9 eV). Moreover, we also consider that it would be unreasonable if the valence state of one Ir atom binding with C atoms is close to 0 (Angew. Chem. Int. Ed. 2022, 61, e202202654; Nat. Chem. 2021, 13, 887-894). As we know, the highly dispersed Ir species bonded with C atoms on ND@G, thereby leading to the partial oxidation of Ir, similar to other metal-based catalysts (Nat. Catal. 2022, 5, 485-493; J. Am. Chem. Soc. 2022, 144, 5108-5115).

Significantly, we have to say that the positively charged Ir^{δ+} species is not equal to IrO_x in our manuscript. In Fourier-transformed *k*³-weighted EXAFS spectra, no obvious Ir-O coordination in other coordination shells demonstrates that there are no IrO₂ clusters or nanoparticles in the Ir₁/ND@G (Figure 2d).

3. For the quasi-*in situ* XAS spectra of reduced catalysts, the samples were reduced in a fixed-bed reactor with 10% H₂/N₂ at corresponding temperature. After it was cooled to room temperature, the reactor was sealed with two globe valves and transferred to the glove box without exposure to air. And then the prepared catalysts were sealed in a measurement plate by Kapton films under Ar protection. The whole process was performed in the glovebox. The reduced sample could not be oxidized by this sample preparation method for XAS data collection. Above experimental details of the XAS experiments have been added in the revised Method.
4. As for the *in-situ* XAS experiments, it is really difficult to detect a high quality signal of single Ir atoms and analyze the precise structure information, limited by the low loading amount (0.02wt% Ir) and the harsh reaction condition (450 °C) in our work. Alternatively, we have used *in-situ* CO-diffuse reflectance infrared Fourier transform spectroscopy (*in-situ* CO-DRIFTS) to accurately identify the structure of Ir₁/ND@G under reaction conditions and reduction conditions.

For *in-situ* CO-DRIFTS of Ir₁/ND@G under reaction conditions, the samples with KBr were placed in the sealed chamber and reduced in 10% H₂ (5 mL/min) for 1 hour at 200°C. After the chamber was heated to 450 °C in He, the gas was switched to reaction gas (2% C₄H₁₀, 2% H₂ He balance) and treated for 1h at reaction condition. And then, the sealed chamber was cooled to 20 °C in reaction gas. CO adsorption was carried out at 20 °C in a flow of 5% CO/He (5 mL/min). Until the surface is completely covered by CO, the gas was switched to He (5 mL/min) and the spectrum was continuously recorded at 20 °C. As shown in **Supplementary Figure 20**, a pair of CO adsorption peaks were apparently observed at 2086 cm⁻¹ and 2017 cm⁻¹ respectively, which could be attributed to dicarbonyl species adsorbing on positively charged Ir species. The two adsorption peaks are consistent with these on the fresh sample. No peaks of atop and bridged CO species on Ir⁰ multi-atomic species have been observed, demonstrating isolated Ir atom bonded with C atoms was stable and reserved its positive valence at the initial reaction step.

Furthermore, in order to evaluate the structure of the Ir₁/ND@G catalyst even in H₂ atmosphere under the reaction temperature, the reduction temperature was raised to 450 °C (reaction temperature). For *in-situ* CO-DRIFTS of Ir₁/ND@G under reduction treatment, the samples with KBr were placed in the sealed chamber and reduced in 10% H₂ (5 mL/min) for 1 hour at 200 °C. The chamber was heated to 450 °C in He after reduction, the gas was switched to 10% H₂ and treated for 1h at 450 °C. And then, the sealed chamber was cooled to 20 °C. CO adsorption was carried out at 20 °C in a flow of 5% CO/He (5 mL/min). Until the surface is completely covered by CO, the gas was switched to He (5 mL/min) and the spectrum was continuously recorded at 20 °C. As shown in **Supplementary Figure 21**, a pair of CO adsorption peaks were apparently observed at 2098 cm⁻¹ and 2046 cm⁻¹ respectively, which could be attributed to dicarbonyl species adsorbing on positively charged Ir species. The two adsorption peaks are consistent with these on the fresh sample. No peaks of atop and bridged CO species on Ir⁰ multi-atomic species have been observed, demonstrating isolated Ir atom bonded with C atoms was stable and reserved its positive valence under reduction treatment.

We appreciate the reviewer for the comment. Following the suggestion, we have added above results and discussion about *in-situ* CO-DRIFTS of Ir₁/ND@G under reaction condition and reduction condition in the revised manuscript (please see Line 250-258 on Page 16-17) and SI (please see **Supplementary Figure 20** and **21**). The experimental details have been added in the revised Method.

Here, we conclude that the Ir₁/ND@G was in the form of isolated Ir atom at the initial step of butane dehydrogenation (BDH). Surely, the single atoms would aggregate into clusters via Ostwald ripening after long-time reduction treatment or dehydrogenation reaction. In our manuscript, the structural evolution from single Ir atoms to clusters leads to depressed conversion rates, as shown in the STEM images of Ir₁/ND@G after 0.5h and 10h BDH. To suppress the influence of possible structural evolution, we verified its atomically dispersed structure after 1 hour BDH and chose the reaction data point at 0.35h (the first point we measured) to calculate the butane conversion and TOF values. We found that single Ir atom has much higher activity than those of Ir clusters and nanoparticles. Therefore, we claim that single Ir atoms can represent Ir₁/ND@G catalyst and Ir-C₃ is a reasonable catalytic model to investigate the reaction mechanism.

Supplementary Figure 20. *in-situ* CO-DRIFTS of Ir₁/ND@G under reaction condition for 1h. Reaction conditions: 450 °C, 2% C₄H₁₀, 2% H₂, He balance.

Supplementary Figure 21. *in-situ* CO-DRIFTS of Ir₁/ND@G under reduction condition for 1h. Reaction conditions: 450 °C, 10% H₂/He.

REVIEWERS' COMMENTS

Reviewer #1 (Remarks to the Author):

I appreciate the comments and references provided by the authors. While I remain skeptical that isolated Ir will remain oxidized under butane dehydrogenation conditions, I'm willing to accept this paper under the following conditions.

1. The full experimental conditions are provided for each experimental method. These need to be as complete as provided in the response letter that describes the XAS procedure. The readers need to know that XPS, STEM, etc., were taken on samples exposed to air, for example. The details of the XAS is needed to convince the reader that these data are good too. Full procedures are needed for others to obtain similar results.
2. The discussion needs to cite and completely discuss the references provided. While Ni is always difficult to fully reduce and is much easier than Pt, Pd, etc., The TOR of single site Ni is lower than that of metallic Ni, but several orders of magnitude. Metallic Ni, however, makes CH₄ and coke. The rate of Ni⁺² is much lower, but is selective for PDH. The cited ACS Catal. 2020, 10, 12932–12942 paper may also likely be metallic Pt under reaction conditions, see for example, the XPS Fig. 3, where both ionic and metallic Pt are present for this single site 0.1 wt% sample. The data at 0.05% was not reported. These authors did not do in situ measurements, nor XAS, so these could very well be metallic, or not. Thus, Pt could be metallic under reaction conditions. For the Nat. Catal. 2022, 5, 1145–1156 paper where Ru in C-N supports, the STEM shows clearly single atoms in the fresh samples and XAS shows only Ru-O bonds. In this paper, however, in situ XAS was obtained. These too had only Ru-O bonds with no metallic phase. The only suspect data is the XANES edge energy which is close to that of metallic Ru, while the white line shows oxidation. I'm not sure why the XANES energy is low, but at least the as received and under reaction EXAFS spectra indicate oxidized Ru under the reaction conditions. Thus, this catalyst appears to remain in an ionic state during reaction.

While this literature shows it is possible to have stable single sites at dehydrogenation conditions, it is not clear that Ir will do this and what makes this possible, when other catalysts readily reduce. In addition, all these references show that metallic NPs are formed at higher loadings. As a result, one needs to be careful about the methods of preparation, support and specific metal.

Despite my reservations, I recommend acceptance of this paper.

REVIEWERS' COMMENTS

Reviewer #1 (Remarks to the Author):

I appreciate the comments and references provided by the authors. While I remain skeptical that isolated Ir will remain oxidized under butane dehydrogenation conditions, I'm willing to accept this paper under the following conditions.

1. The full experimental conditions are provided for each experimental method. These need to be as complete as provided in the response letter that describes the XAS procedure. The readers need to know that XPS, STEM, etc., were taken on samples exposed to air, for example. The details of the XAS is needed to convince the reader that these data are good too. Full procedures are needed for others to obtain similar results.

Response: We thank the review for the suggestion. We carefully check the description of the characterization experiments. Following the suggestion, more detailed description has been added in revised Method.

2. The discussion needs to cite and completely discuss the references provided. While Ni is always difficult to fully reduce and is much easier than Pt, Pd, etc., The TOR of single site Ni is lower than that of metallic Ni, but several orders of magnitude. Metallic Ni, however, makes CH₄ and coke. The rate of Ni⁺² is much lower, but is selective for PDH. The cited ACS Catal. 2020, 10, 12932–12942 paper may also likely be metallic Pt under reaction conditions, see for example, the XPS Fig. 3, where both ionic and metallic Pt are present for this single site 0.1 wt% sample. The data at 0.05% was not reported. These authors did not do in situ measurements, nor XAS, so these could very well be metallic, or not. Thus, Pt could be metallic under reaction conditions. For the Nat. Catal. 2022, 5, 1145–1156 paper where Ru in C-N supports, the STEM shows clearly single atoms in the fresh samples and XAS shows only Ru-O bonds. In this paper, however, in situ XAS was obtained. These too had only Ru-O bonds with no metallic phase. The only suspect data is the XANES edge energy which is close to that of metallic Ru, while the white line shows oxidation. I'm not sure why the XANES energy is low, but at least the as received and under reaction EXAFS spectra indicate oxidized Ru under the reaction conditions. Thus, this catalyst appears to remain in an ionic state during reaction.

Response: We thank the review for the suggestion. The stable single atom catalysts (SACs) of Ni₁/Al₂O₃ and Ru₁/NC are typical examples to verify that isolated metal sites can remain their local structure (include geometric structure and oxidation state) during high-temperature alkane dehydrogenation. These works are significant and influential. Following the suggestion, we discussed and cited these works in the Introduction.

While this literature shows it is possible to have stable single sites at dehydrogenation conditions, it is not clear that Ir will do this and what makes this possible, when other catalysts readily reduce. In addition, all these references show that metallic NPs are formed at higher loadings. As a result, one needs to be careful about the methods of preparation, support and specific metal.

Response: We thank the review for the comment. Due to the existence of Ir-C bonds, the oxidation state of Ir₁/ND@G was near +4. As we know, the fraction of metal-support interface

substantially increased by downsizing supported metal particles. The strong charge transfer between the metal particles and the support renders metal particles with higher electron-deficient charge states, which results in the formation of positively charged metal species

Despite my reservations, I recommend acceptance of this paper.

Response: Many thanks to the Reviewer #1 for continuous efforts, which help us to further improve the manuscript.